# Label-free imaging for quality control of cardiomyocyte differentiation

Tongcheng Qian [1,3 ✉], Tiffany M. Heaster[1,2,3], Angela R. Houghtaling[1], Kexin Sun[1], Kayvan Samimi[1] & Melissa C. Skala [1,2 ✉]

Human pluripotent stem cell (hPSC)-derived cardiomyocytes provide a promising regenerative cell therapy for cardiovascular patients and an important model system to accelerate drug discovery. However, cost-effective and time-efficient platforms must be developed to evaluate the quality of hPSC-derived cardiomyocytes during biomanufacturing. Here, we develop a non-invasive label-free live cell imaging platform to predict the efficiency of hPSC differentiation into cardiomyocytes. Autofluorescence imaging of metabolic co-enzymes is performed under varying differentiation conditions (cell density, concentration of Wnt signaling activator) across five hPSC lines. Live cell autofluorescence imaging and multivariate classification models provide high accuracy to separate low (< 50%) and high (≥ 50%) differentiation efficiency groups (quantified by cTnT expression on day 12) within 1 day after initiating differentiation (area under the receiver operating characteristic curve, 0.91). This non-invasive and label-free method could be used to avoid batch-to-batch and line-to-line variability in cell manufacturing from hPSCs.

[1] Morgridge Institute for Research, Madison, WI, USA. [2] Department of Biomedical Engineering, University of Wisconsin-Madison, Madison, WI, USA. [3]These authors contributed equally: Tongcheng Qian, Tiffany M. Heaster. ✉email: tqian5@wisc.edu; mcskala@wisc.edu

Despite advances in treatment, cardiovascular disease is the leading cause of death worldwide[1]. Globally, about 12% of adults are diagnosed with cardiovascular disease and over 30% of all deaths are caused by cardiovascular disease[1]. The excessive demand of heart transplantation has outpaced the limited number of healthy and functional heart donors[2]. Cell-based regenerative therapy provides a promising treatment for patients suffering from cardiac tissue injury[3,4]. However, cardiomyocytes (CMs) are terminally differentiated cells with no regenerative capacity[5]. Hence, cost-effective and time-efficient platforms to generate functional CMs with high quality has emerged as an urgent need for cardiac medicine in drug screening, toxicity testing, disease modeling, and regenerative cell therapy.

Human pluripotent stem cells (hPSCs) can differentiate into cells from all three germ layers[6–8]. A variety of methods have been established to generate CMs from hPSCs[9–11]. These hPSC-derived CMs exhibit similar functional phenotypes to their in vivo counterparts[11], including self-contractility and action potentials. hPSC-derived CMs have been used in disease modeling[12,13] and drug screening[14], and hold great potential for regenerative medicine[15,16]. However, batch-to-batch and line-to-line variability in the differentiation process of hPSCs into CMs has impeded the scale-up of CM manufacturing[17]. For safety, the quality of clinical-graded hPSC-derived CMs must be rigorously evaluated before they can be used for regenerative cell therapy in patients[18]. Current approaches to quantify CM differentiation rely on low-throughput, labor-intensive, and destructive immunofluorescence labeling and electrophysiological measurements[11]. New technologies that can non-invasively monitor CM differentiation in real time and evaluate the differentiation outcome at early stages are needed to effectively optimize the biomanufacturing of CMs from stem cells.

Previous studies indicate that hPSC-derived CMs undergo dramatic metabolic changes throughout differentiation[19]. Reduced nicotinamide adenine dinucleotide (phosphate) (NAD(P)H) and oxidized flavin adenine dinucleotide (FAD) are autofluorescent cellular metabolic co-enzymes that can be imaged to collect metabolic information at a single-cell level[20]. The ratio of NAD(P)H to FAD intensity is the "optical redox ratio", which reflects the relative oxidation-reduction state of the cell. The fluorescence lifetimes of NAD(P)H and FAD are distinct in the free and protein-bound conformations, so changes in these fluorescence lifetimes reflect changes in protein-binding activity[21,22]. Optical metabolic imaging (OMI) quantifies both NAD(P)H and FAD intensity and lifetime variables. Several groups have demonstrated that autofluorescence imaging can non-invasively track stem cell metabolic activities in real time, including monitoring mesenchymal stem cell differentiation into adipocytes[23,24], osteocytes[24,25], and chondrocytes[25], distinguishing differentiation of hPSCs into dermal and epidermal lineages[26], metabolic difference between hPSCs and feeder cells[27], and hematopoietic stem cells at different stages[28]. These prior studies indicate that OMI is suitable to detect the metabolic changes that occur during CM differentiation.

The goal of this study is to build a predictive model based on OMI to determine whether OMI can predict CM differentiation efficiency early in the differentiation process. Early prediction of CM differentiation outcome can benefit CM manufacturing. We demonstrate a facile method to non-invasively monitor metabolic changes during hPSC differentiation into CMs by combining OMI with quantitative image analysis. OMI is performed at multiple time points during a 12-day differentiation process under varying conditions (cell density, concentration of Wnt signaling activator) and different hPSC lines (human embryonic pluripotent stem cells and human induced pluripotent stem cells).

Differentiation efficiency is quantified by flow cytometry with cTnT labeling on day 12. During the differentiation process all 13 OMI variables, including both NAD(P)H and FAD intensity and lifetime variables, change distinctively between low (< 50% cTnT + on day 12) and high (≥ 50% cTnT+ on day 12) CM differentiation efficiency conditions. Multivariate analysis finds that day 1 cells (24 h after Wnt activation) form a distinct cluster from cells at other time points. Logistic regression models based on OMI variables from cells at day 1 perform well for separating low and high differentiation efficiency conditions with a model performance at 0.91 (receiver operating characteristic (ROC) area under the curve (AUC)). Compared to previous studies[23–28], we specifically contribute a predictive model based on OMI to determine CM differentiation outcome as early as day 1. This label-free and non-destructive method could be used for quality control for CM manufacturing from hPSCs.

## Results

**NAD(P)H and FAD fluorescence change early in the cardiomyocyte differentiation process.** Metabolic state plays an important role in regulating hPSC pluripotency and differentiation[29,30], and can be non-invasively monitored via OMI[20,24]. We recorded the autofluorescence dynamics of NAD(P)H and FAD by OMI during the process of hPSC differentiation into CMs. hPSCs were differentiated following a previous protocol[11], and cells were imaged on differentiation day 0 (immediately pre-treatment with CHIR99021, a Wnt signaling activator), day 1 (24 h post-treatment with CHIR99021), day 3 (immediately pre-treatment with IWP2, a Wnt signaling inhibitor), and day 5 (48 h post-treatment with IWP2). OMI was performed at these time points based on the biphasic role of Wnt signaling activation and inhibition in the CM differentiation protocol[11] (Supplementary Fig. 1a). On differentiation day 12, CM differentiation efficiencies were evaluated by flow cytometry with a cardiac specific marker cTnT. Differentiation of CMs from hPSCs critically relies on the timing and the state of Wnt signaling[11]. Both the concentration of CHIR99021[31] and cell density[7] are closely related to the activation level of the Wnt signaling pathway. In the current study, CM differentiation efficiencies ranging from nearly 0 to above 60% were achieved by initiating CM differentiation with different CHIR99021 concentrations and hPSC seeding densities (Fig. 1a, b, Table 1).

A total of 13 OMI variables, including the optical redox ratio, NAD(P)H intensity and lifetime variables ($\tau_1$, $\tau_2$, $\alpha_1$, $\alpha_2$, $\tau_m$), FAD intensity and lifetime variables ($\tau_1$, $\tau_2$, $\alpha_1$, $\alpha_2$, $\tau_m$) were measured by autofluorescence imaging. The short lifetime ($\tau_1$) corresponds to free NAD(P)H while the long lifetime ($\tau_2$) corresponds to protein-bound NAD(P)H. The converse applies to FAD $\tau_1$ (protein-bound) and $\tau_2$ (free). Weights are applied to the short ($\alpha_1$) and long ($\alpha_2$) lifetimes, and the mean lifetime is a weighted average ($\tau_m = \alpha_1\tau_1 + \alpha_2\tau_2$). Cells under the lowest differentiation efficiency condition (0.3%, Table 1) and highest differentiation efficiency condition (65.5%, Table 1) showed significant differences in OMI variables by day 1. Cells with the highest differentiation efficiency had a lower FAD $\tau_m$ on day 0 and a higher FAD $\tau_m$ on day 1 compared to the lowest differentiation efficiency at the same time points (Fig. 1c–e). Similarly, the fold change between day 0 and day 1 for NAD(P)H $\tau_m$ (Fig. 1f–h) and the optical redox ratio (Fig. 1i–k) is greater for high differentiation efficiency compared to low differentiation efficiency conditions. Significant differences in other OMI variables were observed between day 0 and day 1, as well as between low and high differentiation efficiency conditions (Supplementary Fig. 1b). After treating H9 embryonic stem cells with an inhibitor of glycolysis (2-DG)[32], the optical redox ratio changed oppositely

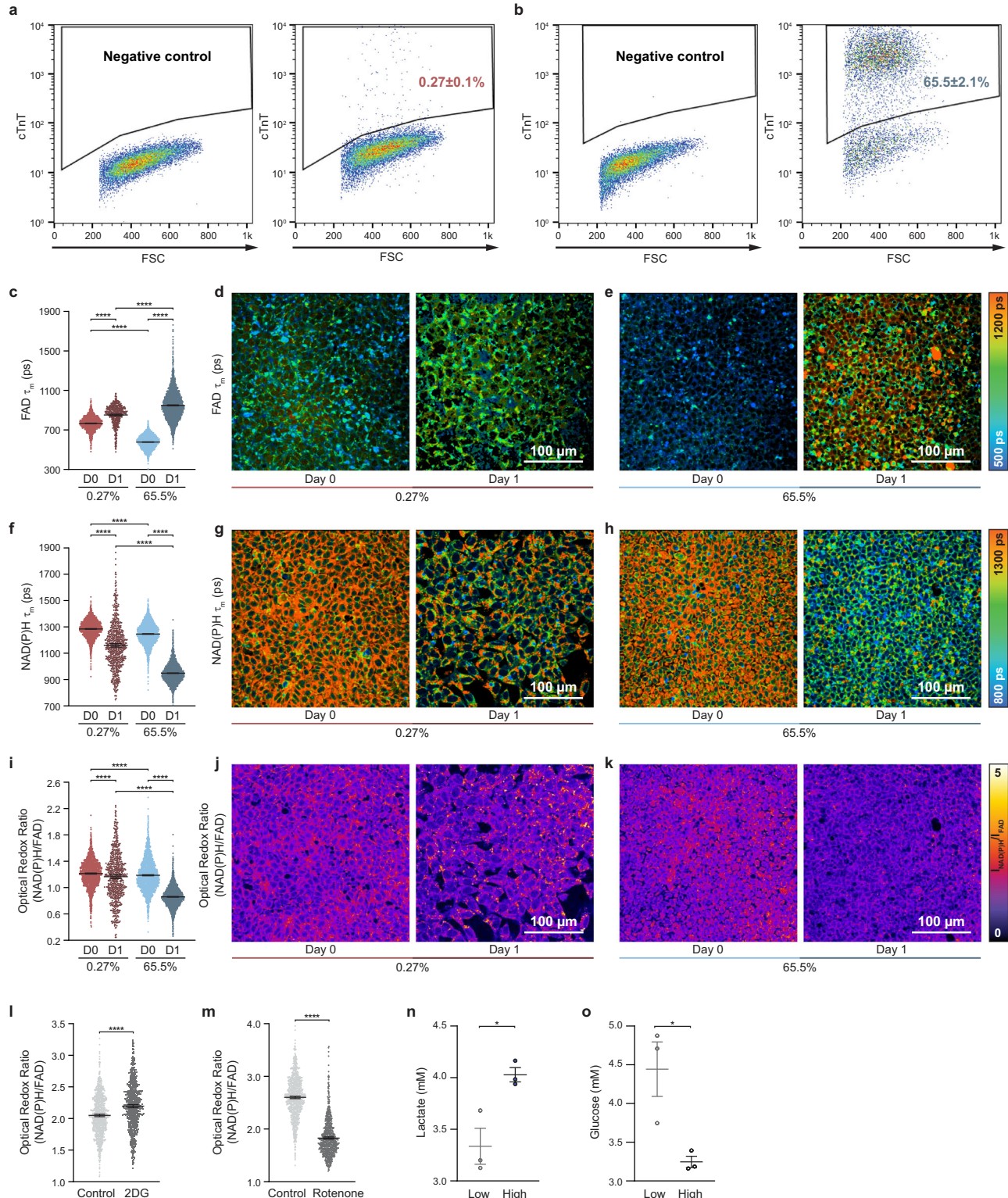

compared to hPSCs undergoing CM differentiation in the first 24 h (Fig. 1i–l, Supplementary Fig. 1b). However, the optical redox ratio decreased both in H9 embryonic stem cells after rotenone treatment (an oxidative phosphorylation inhibitor)[33] (Fig. 1m) and in hPSCs undergoing CM differentiation in the first 24 h (Fig. 1i–k). Lactate and glucose concentrations in the cell culture medium after 24 h of differentiation for low and high differentiation efficiency conditions revealed more glycolytic activity in the high differentiation efficiency condition (Fig. 1n, o). Changes

in autofluorescence with known metabolic inhibitors and during CM differentiation together with lactate and glucose assay results indicate that differentiating cells altered their metabolic activity 1 day after CHIR99021 treatment. This observation is consistent with previous studies that found metabolism differed between hPSCs and differentiated cells, and between cells differentiated into CMs and other cell types[34]. Overall, auto-fluorescence imaging of NAD(P)H and FAD showed significant changes at early time points in the differentiation process,

**Fig. 1 NAD(P)H and FAD fluorescence change differently and reflect different glycolytic activity in the first 24 h for cells in low vs. high cardiomyocyte differentiation efficiency conditions.** hPSCs were differentiated into CMs following an established method[11]. On differentiation day 12, cells were verified by flow cytometry with cTnT labeling from three independent replicates. **a, b** Representative flow cytometry dot plots for **a** low and **b** high differentiation efficiencies along with negative controls. Gating strategy to determine the percentage of cTnT positive population in hPSC-derived cells. Single-cell quantitative analysis of mean lifetimes ($\tau_m$, reported as picoseconds) of **c–e** FAD and **f–h** NAD(P)H, and **i–k** optical redox ratio for low (0.3% cTnT+) and high differentiation (65.5% cTnT+) efficiencies on day 0 ("D0", immediately pre-treatment) and day 1 ("D1", 24 h post-treatment with CHIR99021), and their corresponding representative images. $n = 2458$, 633, 3534, and 4446 cells for 0.3% day 0, 0.3% day 1, 65.5% day 0, and 65.5% day 1, respectively. Data are presented as dot plots with bars for the mean and 95% CI for each condition each day. Statistical significance was determined by one-way analysis of variance (ANOVA) followed by Tukey's post hoc tests. ****$p < 0.0001$. Color bars are indicated on the right. Changes of optical redox ratio after treatment with 2DG or rotenone. **l** Single-cell quantitative analysis of optical redox ratio for H9 ESCs before and 2 h after 10 mM 2DG treatment. $n = 1051$ and 900 cells for before and after 2DG treatment, respectively. **m** Single-cell quantitative analysis of optical redox ratio for H9 ESCs before and 15 min after 10 μM rotenone treatment. $n = 1042$ and 986 cells for before and after rotenone treatment, respectively. Data are presented as dot plots with bars for the mean and 95% CI. Statistical significance was determined by unpaired two-tailed Student's $T$-test. ****$p < 0.0001$. ps, picoseconds. After the first 24 h of differentiation, **n** lactate and **o** glucose concentrations of cell culture medium from low (10.8%) and high (63.1%) differentiation efficiency conditions were measured with three biological replicates, respectively. Data are presented as dot plots with mean ± SEM. Statistical significance was determined by unpaired two-tailed Student's $T$-test. *$p = 0.0210$ and 0.0291 for **n** lactate assay and **o** glucose assay, respectively. Source data are provided as a source data file.

**Table 1 Summary of the 15 differentiation conditions.**

| Condition | hPSC line | Seeding density (cells/well) | CHIR99021 concentration | Differentiation efficiency |
|---|---|---|---|---|
| 1.5 m–12 μM | H9 ESC | $1.5 \times 10^6$ | 12 μM | *0.3 ± 0.1%* |
| 1.5 m–9 μM | H9 ESC | $1.5 \times 10^6$ | 9 μM | *0.5 ± 0.2%* |
| 100 k–12 μM | H9 ESC | $1.0 \times 10^5$ | 12 μM | *0.6 ± 0.3%* |
| 19-9-11-600 k–8 μM | 19-9-11 iPSC | $6.0 \times 10^5$ | 8 μM | *5.9 ± 0.7%* |
| H13-600 k–10 μM | H13 ESC | $6.0 \times 10^5$ | 8 μM | *10.8 ± 1.3%* |
| 2 m–12 μM | H9 ESC | $2.0 \times 10^6$ | 12 μM | *15.1 ± 1.3%* |
| 500 k–12 μM | H9 ESC | $5.0 \times 10^5$ | 12 μM | *19.6 ± 1.5%* |
| 1 m–12 μM | H9 ESC | $1.0 \times 10^6$ | 12 μM | *21.7 ± 2.3%* |
| IMR90-1 m–12 μM | IMR90-4 iPSC | $1.0 \times 10^6$ | 12 μM | *26.4 ± 1.2%* |
| IMR90-1 m–9 μM | IMR90-4 iPSC | $1.0 \times 10^6$ | 9 μM | *38.5 ± 1.9%* |
| 500 K–10 μM | H9 ESC | $5.0 \times 10^5$ | 10 μM | **51.8 ± 3.2%** |
| IMR90-1 m–10 μM | IMR90-4 iPSC | $1.0 \times 10^6$ | 10 μM | **53.2 ± 0.8%** |
| 19-9-11-600 k–6 μM | 19-9-11 iPSC | $6.0 \times 10^5$ | 6 μM | **61.0 ± 3.2%** |
| H13-800 k–10 μM | H13 ESC | $8.0 \times 10^5$ | 10 μM | **63.1 ± 2.4%** |
| 500 K–9 μM | H9 ESC | $5.0 \times 10^5$ | 9 μM | **65.5 ± 2.1%** |

hPSCs, including H9 and H13 embryonic stem cells (ESC) or IMR90-4 and 19-9-11 induced pluripotent stem cells (iPSC) were differentiated into CMs following an established method[11]. H13 ESC, IMR904- iPSC, and 19-9-11 iPSC are specified with H13, IMR90, and 19-9-11. On differentiation day 12, cells were verified by flow cytometry with cTnT labeling from three independent replicates to define differentiation efficiency. Data were collected from three biological replicates. Conditions are presented with condition name (seeding density, CHIR99021 concentration, IMR90 status), hPSC line, seeding density, CHIR99021 (Wnt activator) concentration, and differentiation efficiency (mean ± SEM). Low differentiation efficiencies (< 50% cTnT+ on day 12) are in italic and high differentiation efficiencies (≥ 50% cTnT+ on day 12) are in bold.

with greater changes in higher CM differentiation efficiency conditions.

**Multivariate analysis reveals unique NAD(P)H and FAD fluorescence in cells 1 day into the differentiation process.** To assess differences in OMI variables across days, cells were clustered across all days (day 0, day 1, day 3, and day 5) and differentiation conditions (Table 1) with a Uniform Manifold Approximation and Projection (UMAP) dimension reduction technique[35]. UMAP dimensionality reduction was performed on all 13 OMI variables for projection onto 2D space. UMAP representations of all OMI variables showed a day 1 subpopulation separated from days 0, 3, 5 (Fig. 2a, Supplementary Fig. 2). CM differentiation efficiency conditions were separately evaluated across all days by UMAP. As shown in Fig. 2c, day 1 cells (light blue clusters) from high (≥ 50%) differentiation efficiency conditions were distinctly clustered, while cells from low (< 50%) differentiation efficiency conditions clustered together across all days. Therefore, differentiation conditions were separated into low differentiation efficiency (< 50% cTnT+ on day 12, Table 1 in italic) and high differentiation efficiency (≥ 50% cTnT+ on day 12, Table 1 in bold).

Heatmap dendrogram clustering based on OMI variable z-scores revealed that cells under high differentiation efficiency conditions on day 1 were clustered closely together and distinct from cells under low differentiation efficiency conditions on day 1 (Fig. 2b) except the 61.0% differentiation efficiency with 6 μM CHIR treatment for 19-9-11 hPSCs. This outlier could be due to multiple reasons, e.g., the low CHIR99021 concentration or low cell seeding density. In the future, more variables, such as cell morphology, can be introduced to increase linear clustering performance. Dendrograms of cells on day 0 and day 1 together (Supplementary Fig. 3a) or day 0 alone (Supplementary Fig. 3b) did not show clear separation of high and low differentiation efficiency conditions, indicating that day 1 is the earliest time point to separate low and high differentiation efficiency conditions. In summary, UMAP clustering of all 13 OMI variables across all time points and z-score heatmap clustering from day 0 and day 1 across all differentiation conditions showed that cells under high differentiation efficiency conditions on day 1 clustered together and were distinct from other conditions and time points. Hence, we hypothesize that OMI of live cells on CM differentiation day 1 could predict high and low differentiation efficiencies on day 12.

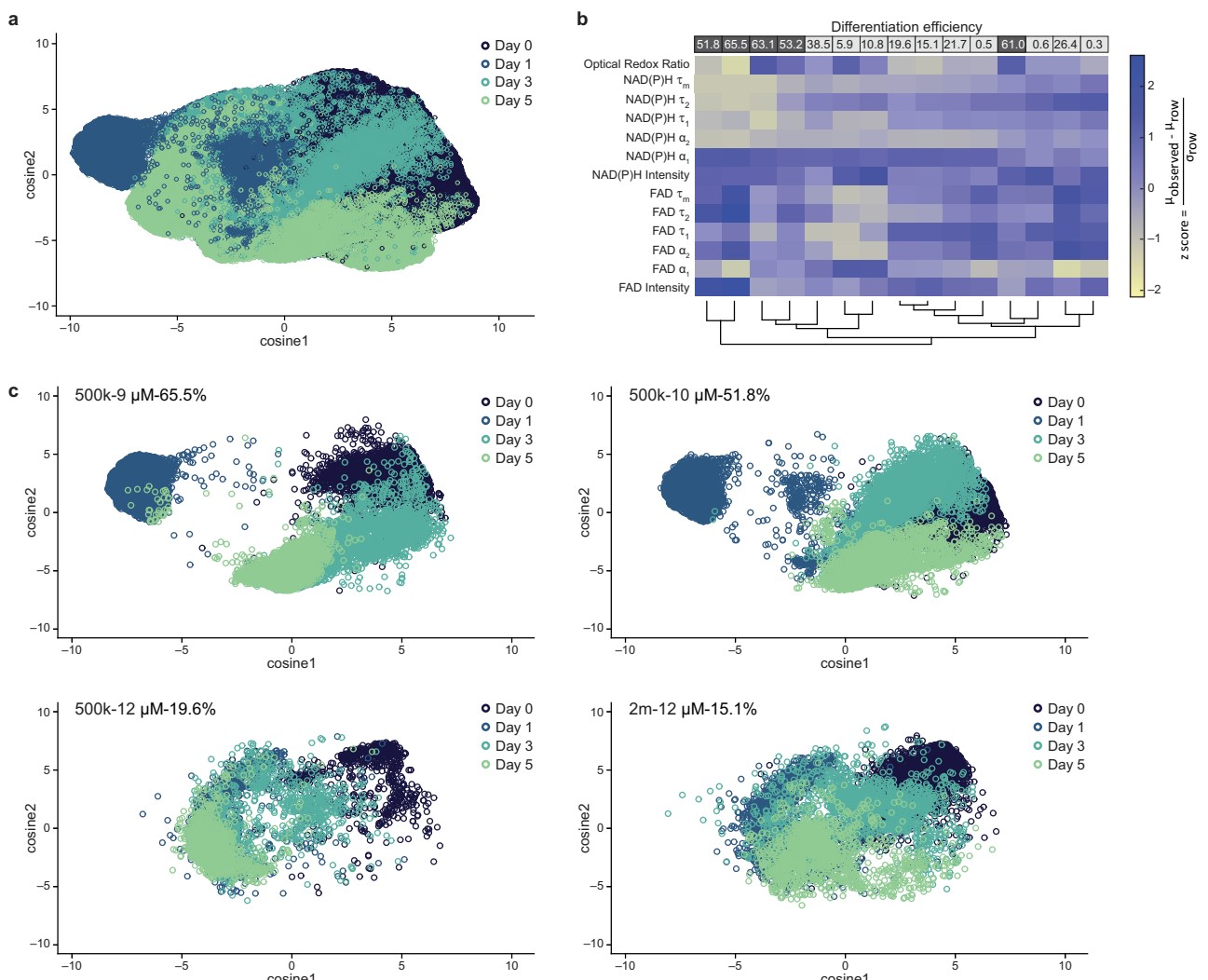

**Fig. 2 Multivariate analysis reveals unique metabolic profiles in cells differentiated into cardiomyocytes at day 1. a** Uniform Manifold Approximation and Projection (UMAP) dimensionality reduction was performed on all 13 autofluorescence variables (optical redox ratio, NAD(P)H $\tau_m$, $\tau_1$, $\tau_2$, $\alpha_1$, $\alpha_2$, and intensity; FAD $\tau_m$, $\tau_1$, $\tau_2$, $\alpha_1$, $\alpha_2$, and intensity) for each cell and projected onto 2D space. Cells from all 11 conditions shown in Table 1 are plotted together. Data include cells from day 0, day 1, day 3, and day 5. Each dot represents one single cell, and $n = 25,304$, $25,470$, $26,228$, and $23,484$ cells for day 0, 1, 3, and 5, respectively. **b** Heatmap dendrogram clustering based on similarity of average Euclidean distances across all variable z-scores was performed on day 1 cells across all 15 conditions. Conditions are indicated by the CM differentiation efficiency percentages as noted by column labels at the top of the heatmap (quantified by flow cytometry cTnT+ on day 12, full conditions given in Table 1). Low differentiation efficiencies (< 50%) are in italic and high differentiation efficiencies are in bold (≥ 50%). Z-score $= \frac{\mu_{observed} - \mu_{row}}{\sigma_{row}}$, where $\mu_{observed}$ is the mean value of each variable for each condition; $\mu_{row}$ is the mean value of each variable for all 15 conditions together, and $\sigma_{row}$ is the standard deviation of each variable across all 15 conditions. Autofluorescence variables are indicated on the left side as row labels. $n = 30463$ cells from day 1. **c** Separated UMAP clusters for representative differentiation conditions. Conditions are labeled with original cell seeding density, CHIR99021 treatment concentration, and final cardiomyocyte differentiation efficiency quantified by flow cytometry (detailed in Table 1). $n = 13,897$, $13,852$, $4357$, and $7601$ cells for condition 65.5%, 51.8%, 19.6%, 15.1%, respectively. Source data are provided as a source data file.

**OMI variables accurately distinguish cells under low or high differentiation efficiency conditions on day 1.** After identifying distinct clustering of day 1 cells in high differentiation efficiency conditions based on all 13 OMI variables, we further explored day 1 OMI data alone. Cells in high differentiation efficiency conditions (Fig. 3a, dark gray, ≥ 50% cTnT+ on day 12) formed a distinct cluster from cells under low differentiation efficiency conditions (Fig. 3a, light gray, < 50% cTnT+ on day 12) on day 1. However, a small portion of cells from high and low differentiation efficiency conditions overlap. Note that the high differentiation efficiency conditions were not 100% and the low differentiation efficiency conditions were not 0%, so this could explain some of the overlap on day 1.

Next, day 1 data were separated into two datasets for classification training (dataset 1), validation (dataset 1), and prediction (dataset 2), where dataset 1 and 2 are independent biological replicates. A logistic regression classifier based on all 13 OMI variables was trained on 80% of day 1 single cell dataset 1 to classify cells from low vs. high differentiation efficiency conditions and validated on the remaining 20% of day 1 single cell dataset 1. Reported variable weights and accuracy scores were validated from dataset 1. Performance of classifiers (receiver operating characteristic (ROC) area under the curve (AUC)) was evaluated on dataset 2. Variable weights indicated that NAD(P)H $\tau_m$ and FAD $\tau_m$ were important variables for discriminating low vs. high differentiation efficiency conditions (Fig. 3b). Logistic

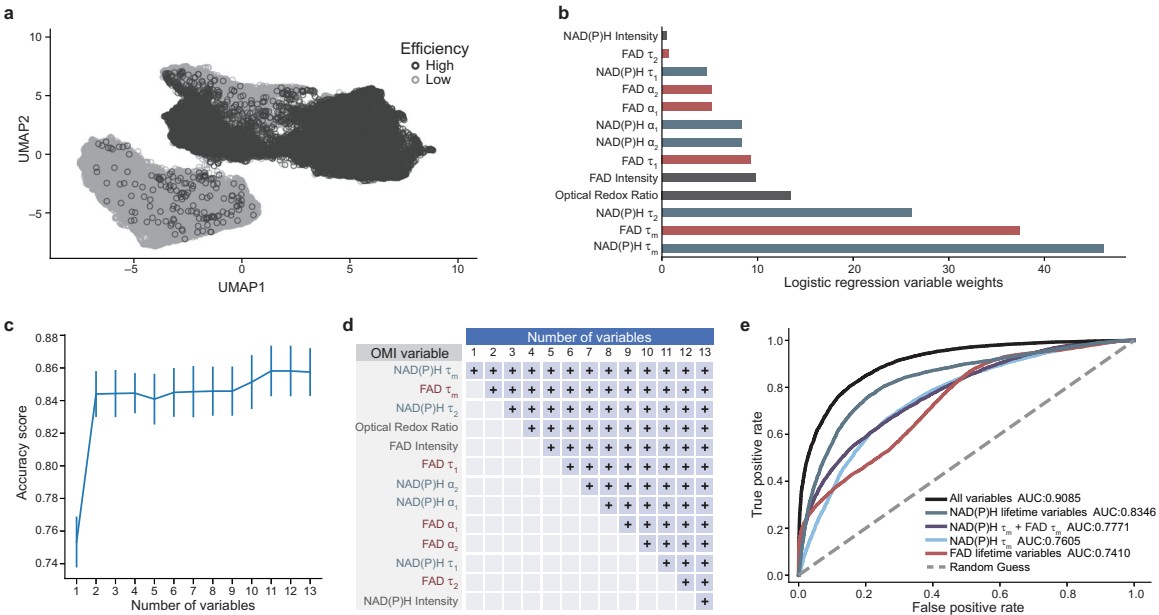

**Fig. 3 OMI variables accurately distinguish cells under low or high cardiomyocyte differentiation efficiency conditions on day 1. a** UMAP dimensionality reduction was performed on all 13 OMI variables (optical redox ratio, NAD(P)H $\tau_m$, $\tau_1$, $\tau_2$, $\alpha_1$, $\alpha_2$, and intensity; FAD $\tau_m$, $\tau_1$, $\tau_2$, $\alpha_1$, $\alpha_2$, and intensity) for each cell on day 1 and projected onto 2D space. Day 1 cells from all 15 conditions shown in Table 1 are plotted together with cells from low (< 50% cTnT+ on day 12) and high (≥ 50% cTnT+ on day 12) CM differentiation efficiencies in light gray and dark gray, respectively. $n = 16048$ and 14415 cells for low and high differentiation efficiency conditions, respectively. **b–e** All OMI data from day 1 cells were separated into two datasets. Dataset 1 was randomly partitioned into 80% portions for training and 20% portions for testing, respectively ($n = 8974$ cells for training, $n = 2244$ cells for test). Dataset 2 was used for evaluation of classifier performance. Binary classification was tested for low (< 50% cTnT+) vs. high (≥ 50% cTnT+) differentiation efficiency conditions on day 1. **b** OMI variable weights are shown specific to the logistic regression model. **c** Classification accuracy with respect to number of OMI variables was evaluated by chi-squared variable selection to separate low and high differentiation efficiency conditions with the logistic regression model. The number of variables included in the logistic regression model are indicated at bottom-axis. The accuracy scores are presented as mean ± STDEV. **d** The variables included for each logistic regression model [specified by numbers of variables on the x-axis in (**c**)] are defined, where the blue text indicates NAD(P)H lifetime variables and the red text indicates FAD lifetime variables. The OMI variables included in each instance (e.g., 3, 4) are indicated by a light blue + in each column. **e** Model performance of the logistic regression classifier was evaluated by receiver operating characteristic (ROC) curves using different OMI variable combinations as labeled. The area under the curve (AUC) is provided for each variable combination as indicated in the legend. Source data are provided as a source data file.

regression (Fig. 3c, d), support vector machine (Supplementary Fig. 4a, b), and random forest (Supplementary Fig. 4c, d) classifiers were generated to test the prediction accuracy using all 13 OMI variables, yielding an accuracy score > 85% for all three classifiers. ROCs based on logistic regression classifiers using all 13 OMI variables and a subset of variables are shown in Fig. 3e along with their performance defined by the AUC. Here, the logistic regression classifier using all 13 OMI variables achieves an AUC > 0.90 (Fig. 3e). With only NAD(P)H lifetime variables (NAD(P)H $\tau_m$, $\tau_2$, $\alpha_2$, $\alpha_1$, $\tau_1$) that can be collected in the NAD(P)H channel alone, the AUC is > 0.83 (Fig. 3e). The AUC with NAD(P)H lifetime variables is higher than the AUC of other variable subsets, including FAD lifetime variables, NAD(P)H $\tau_m$ and FAD $\tau_m$ together, and NAD(P)H $\tau_m$ alone (Fig. 3e). This model indicates that NAD(P)H lifetime variables alone perform better than the combination of NAD(P)H and FAD $\tau_m$ (Fig. 3e). Given that all NAD(P)H lifetime variables must be measured to calculate NAD(P)H $\tau_m$ and that additional laser lines are needed to add FAD measurements, we conclude that NAD(P)H lifetime measurements provide a balance of good accuracy with reduced complexity. However, all NAD(P)H and FAD intensity and lifetime measurements provide the highest accuracy. Hence, NAD(P)H lifetime variables alone are sufficient to predict low vs. high CM differentiation efficiency conditions. Additionally, support vector machine and random forest classifiers using all 13 OMI variables achieve an AUC > 0.90 (Supplementary Fig. 4e). These data indicate that OMI can accurately predict CM differentiation

under low vs. high differentiation efficiency conditions at an early time point (day 1).

**Imaging of a cardiac reporter line confirms autofluorescence changes in cells under high differentiation efficiency conditions.** Given that OMI can identify CM differentiation efficiency at an early stage, we evaluated a CM reporter line (NKX2.5[EGFP/+] hPSCs)[36] to track differentiated CMs together with autofluorescence imaging during the entire differentiation process. The NKX2.5[EGFP/+] hPSC line expresses EGFP when the cardiac progenitor protein NKX2.5 is expressed, indicating that the cell has differentiated into CM, around differentiation day 7. Although EGFP spectrally overlaps with FAD autofluorescence signals, this interference does not occur until day seven[36]. The final differentiation efficiency was quantified by flow cytometry with cTnT labeling (Fig. 4a). Approximately 0.3% and 84.1% CMs were yielded with 12 μM and 9 μM CHIR99021 treatment, respectively.

Differences in OMI variables between low (0.3%) and high (84.1%) differentiation efficiencies were assessed with this reporter line. Consistent with the observations in Fig. 1c and d, NAD(P)H $\tau_m$ (Fig. 4b–d) and FAD $\tau_m$ (Fig. 4e–g) were significantly different between low vs. high differentiation efficiency conditions on day 0 and day 1 with the CM reporter line. In both conditions, NAD(P)H $\tau_m$ gradually decreased over the first 5 days of differentiation (Supplementary Fig. 5a). Conversely, FAD $\tau_m$ gradually increased over the first 5 days

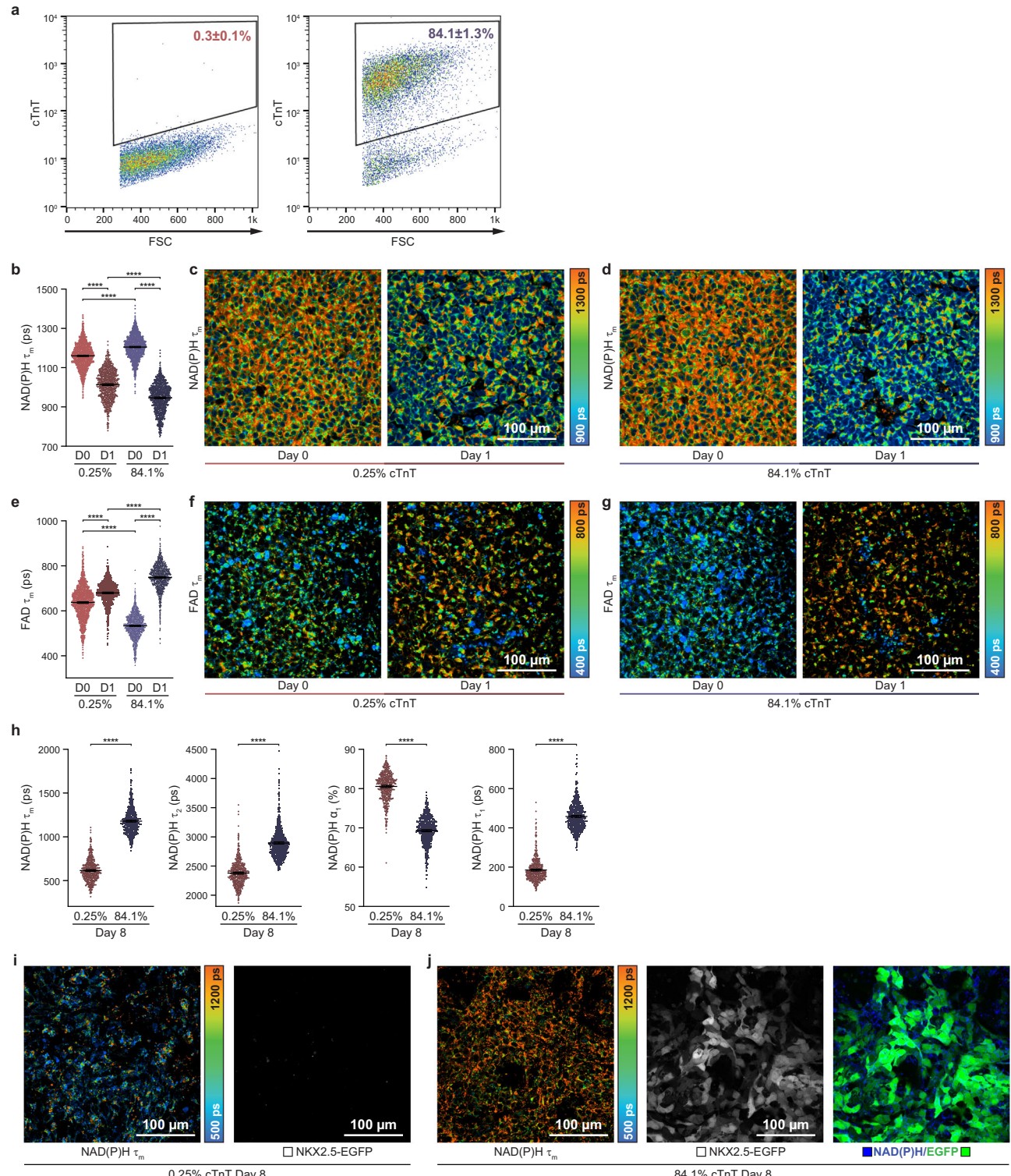

for the high differentiation efficiency condition and oscillated over time for the low differentiation efficiency condition (Supplementary Fig. 5b). These observations are consistent with our previous findings (Fig. 1, Supplementary Fig. 1b) using 11 differentiation conditions across two hPSC lines.

After confirming that EGFP lifetimes are consistent with previous measurements[37] and do not overlap with NAD(P)H lifetimes (Supplementary Fig. 6), we then evaluated the NAD(P)H lifetimes of differentiated CMs on day 8 when the cells expressed NKX2.5-EGFP. As shown in Fig. 4h–j, NAD(P)H lifetime

variables ($\tau_m$, $\tau_2$, $\alpha_1$, $\tau_1$) were significantly different between low and high differentiation efficiencies on day 8. OMI differences between CMs and non-CMs on day 8 further confirmed that autofluorescence can identify CMs at different stages during differentiation. Similar changes in NAD(P)H lifetime variables were also observed in H9 embryonic stem cells treated with an inhibitor of glycolysis (2-DG)[32] (Supplementary Fig. 7). In summary, with a cardiac reporter line, we further confirmed that NAD(P)H and FAD fluorescence variables reflect CM differentiation efficiency from hPSCs. Differentiated CMs (84.1%) exhibit

**Fig. 4 Imaging of a cardiac reporter line confirms autofluorescence changes in cells under high differentiation efficiency conditions.** NKX2.5[EGFP/+] hPSCs were treated with 12 μM and 9 μM of CHIR99021 for the first 24 h to achieve low and high differentiation efficiencies, respectively. The NKX2.5[EGFP/+] hPSC line expresses EGFP when the cardiac progenitor protein NKX2.5 is expressed, indicating that the cell has differentiated into CM, around differentiation day 7. **a** CM differentiation efficiencies were verified by flow cytometry on day 12 with cTnT labeling. Low differentiation efficiency (12 μM CHIR99021, left) and high differentiation efficiency (9 μM CHIR99021, right) are shown. Data were collected from three biological replicates and presented as mean ± SEM. Gating strategy to determine the percentage of cTnT positive population in hPSC-derived cells. Single-cell quantitative analysis of **b** NAD (P)H mean lifetimes ($\tau_m$), **c, d** representative images, and **e** FAD mean lifetimes ($\tau_m$), **f, g** representative images for low (0.3% cTnT+ on day 12) and high (84.1% cTnT+ on day 12) differentiation efficiencies on day 0 ("D0", immediately pre-treatment) and day 1 ("D1", 24 h post-treatment with CHIR99021), respectively. $n = 1618, 1017$ cells for 0.3% condition day 0; day 1, respectively. $n = 1633, 1243$ cells for 84.1% condition day 0; day 1, respectively. Data are presented as dot plots with bars for the mean and 95% CI. Statistical significance was determined by one-way analysis of variance (ANOVA) followed by Tukey's post hoc tests. ****$p < 0.0001$. **h** Single-cell quantitative analysis of NAD(P)H $\tau_m$, $\tau_2$, $\alpha_1$, $\tau_1$ on differentiation day 8 (Differentiation efficiencies are indicated at the bottom as percent cTnT+ on day 12). Statistical significance was determined by Student's T-test. ****$p < 0.0001$. $n = 580$ and $727$ cells for 0.3% and 84.1% condition day 8, respectively. Data are presented as dot plots with bars for the mean and 95% CI. Representative images NAD(P)H $\tau_m$ and EGFP fluorescence in live cells for **i** low differentiation efficiency (0.3% cTnT+) and **j** high differentiation efficiency (84.1% cTnT+). ps, picoseconds. Source data are provided as a source data file.

dramatically different autofluorescence compared to differentiated non-CMs (0.3%), which provides further evidence that OMI can discriminate between CMs and non-CMs after the differentiation process is complete.

## Discussion

Here, we report a non-invasive label-free imaging method to predict the outcome of hPSC differentiation into CMs. By combining live cell autofluorescence lifetime imaging, single-cell image analysis, and machine learning, we robustly separate low (< 50%) from high (≥ 50%) CM differentiation efficiency conditions as early as day 1. The prediction accuracy was over 85% and the model performance was 0.91 (AUC of ROC) with all 13 OMI variables combined across 15 different differentiation conditions including 5 hPSC lines.

Recent evidence links Wnt signaling and glycolytic activities during hPSC differentiation into mesoderm[38,39]. Consistent with these findings, changes in OMI variables on day 1 of CM differentiation (24 h after Wnt activation, Fig. 1) and with known metabolic inhibitors in stem cells together with lactate and glucose assays (Fig. 1l–o, Supplementary Fig. 7) indicate that changes in OMI variables are due to increased glycolytic activity on day 1 of CM differentiation. Considering the important role of Wnt signaling activation in mesoderm and CM differentiation[11], and embryonic development[40], greater changes in OMI variables in the high differentiation efficiency condition could indicate more glycolytic activity due to successful Wnt activation compared to the low differentiation efficiency condition (Fig. 1c–k, Supplementary Fig. 1). Taken together, these results reveal that autofluorescence imaging can separate CM differentiation efficiencies at an early stage based on metabolic changes (Fig. 3b–e).

At the end of our differentiation process, some cells were not positive for cTnT and therefore were not CMs. Previous studies have shown that these non-CMs at the end of the differentiation process are primary cardiac-like fibroblasts together with a small portion of non-differentiated hPSCs[41]. hPSC-derived CMs exhibit distinct metabolism from other hPSC-derived non-CMs[42]. Co-culture of cardiac fibroblasts and CMs can induce fibroblast glycolytic activity and lactate secretion from fibroblasts[43]. Co-culture of cardiac fibroblasts and CMs also promotes a more mature phenotype in CMs along with increased reliance on oxidative phosphorylation[41]. Similarly, differences in NAD(P)H lifetime variables between CMs and non-CMs on day 8 (Fig. 4h) are consistent with decreased glycolytic activity in the CMs (Supplementary Fig. 7). These results further confirm that autofluorescence imaging can distinguish the distinct metabolic activities between hPSC-derived CMs and other non-CMs with the NKX2.5-EGFP reporter line. OMI distinguished different cell populations at multiple time-points in this differentiation protocol, with larger differences between low and high differentiation efficiencies on day 8 than on day 1. Day 1 cells are mainly primitive streak whereas cells on day 8 are at the end of differentiation and exhibit the glycolytic activities of fetal/newborn CMs[44]. Therefore, larger differences are expected in OMI between CMs and non-CMs at day 8 compared to low and high differentiation efficiencies on day 1. OMI differences between low and high differentiation efficiencies at multiple days in this protocol indicates that this technology could continuously monitor stem cell differentiation stages.

We have demonstrated that autofluorescence imaging can resolve metabolic changes in CM differentiation and predict the differentiation outcome at early time points. However, our method has limitations. The differentiation efficiency of hPSCs is susceptible to cell line variability, cell culture microenvironment, and differentiation protocol[45]. We note that the differentiation efficiency measured from flow cytometry in our experiments was not higher than 90%. This may be due to photo-toxicity during the imaging process that may moderately interrupt CM differentiation. In future studies, good manufacturing practice standards could be applied to optimize the evaluation process and minimize the interruption on differentiation. Additionally, OMI relies on only two metabolites, NAD(P)H and FAD, that do not comprehensively characterize cellular metabolic activities. More mechanistic studies together with other assays, including metabolite liquid chromatography–mass spectrometry[29], NMR spectrometry[46], single-cell RNASeq[47], and quantitative proteomics[48], need to be performed to reveal the relationship between metabolic dynamics and hPSC differentiation into CMs. Additionally, alternative differentiation protocols will require algorithms trained on OMI data in these new conditions to robustly classify differentiation efficiencies.

Overall, we developed a non-invasive method to predict the efficiency of hPSC differentiation into CMs at early differentiation stages. hPSCs hold great promise for regenerative medicine and pharmaceutical development, but large-scale cell manufacturing suffers from variability across hPSC lines and cell culture conditions. Our studies indicate that autofluorescence can predict CM differentiation efficiency at an early stage, which could enable real-time and/or in-line monitoring during cell manufacturing. This method could lower manufacturing costs and personnel time by flagging samples for timely interventions. Similar technologies could also impact other areas of regenerative cell manufacturing. A few studies have demonstrated that intravital autofluorescence imaging can monitor cellular metabolic activities, including tumor treatment resposne[49] and heterogeneity of tumor tissues[50]. Therefore, the methods described in the current study could be

used to track stem cell differentiation in vivo, with applications in label-free assessment of stem cell-generated tissue integration for in vivo tissue repair.

## Methods

**hPSC culture and cardiomyocyte differentiation.** Human H9 and H13 embryonic stem cells[51], human IMR90-4 and 19-9-11 induced pluripotent stem cells[52], and NKX2.5$^{EGFP/+}$ hPSCs[36] were maintained on Matrigel (Corning)-coated surfaces in mTeSR1 (STEMCELL Technologies) as previously described[53]. CM differentiation was performed as described previously[11]. A step-by-step protocol describing the differentiation method can be found at Protocol Exchange (DOI: 10.21203/rs.3.pex-1571/v1). Different cell seeding densities and different concentrations of CHIR99021 were applied to manipulate differentiation efficiency. Briefly, hPSCs were singularized with Accutase (Thermo Fisher Scientific) and plated onto Matrigel-coated plates at a density ranging from of $2.9 \times 10^4$ cells/cm$^2$ to $5.7 \times 10^5$ cells/cm$^2$ ($1.0 \times 10^5$ cells to $2.0 \times 10^6$ cells per well of a 12-well plate) in mTeSR1 supplemented with 10 μM Rho-associated protein kinase (ROCK) inhibitor Y-27632 (Selleckchem) 2 days before initiating differentiation. Differentiation was initiated by Wnt signaling activation with 8–12 μM CHIR99021 (Selleckchem) on day 0, followed by inhibition of Wnt signaling with 5 μM IPW2 on day 3.

**Flow cytometry.** Cells on differentiation day 12 were disassociated with Accutase, fixed in 1% PFA for 15 min at room temperature, and then blocked with 0.5% bovine serum albumin (BSA) with 0.1% Triton X-100. Cells were then stained with primary antibody anti-cTnT (Lab Vision; 1:200) and secondary antibody (Thermo Fisher; goat anti-mouse, Alexa Fluor 488; 1:500) in 0.5% BSA with 0.1% Triton X-100. Data were collected on a FACSCalibur flow cytometer and analyzed with FlowJo. Data were collected from three biological replicates and presented as means ± SEM. cTnT positive percentage was rounded up at one decimal place.

**Lactate and glucose assays.** After the first 24 h of differentiation, cell culture media were collected from three biological replicates for low and high differentiation efficiency conditions, respectively. Cell culture media were used for lactate and glucose assays following the kit instructions (Biovision, Lactate colorimetric assay kit and Glucose colorimetric assay kit). Absorbance was read by a plate reader at 570 nm.

**Autofluorescence imaging of NAD(P)H, FAD, and NKX2.5-EGFP.** Fluorescence lifetime imaging (FLIM) was performed by an Ultima two-photon imaging system (Bruker) composed of an ultrafast tunable laser source (Insight DS+, Spectra Physics) coupled to a Nikon Ti-E inverted microscope with time-correlated single photon counting electronics (SPC-150, Becker & Hickl, Berlin, Germany). The ultrafast tunable laser source enables sequential excitation of NAD(P)H at 750 nm and FAD at 890 nm. NAD(P)H and FAD emission was isolated using 440/80 nm and 550/100 nm bandpass filters (Chroma), respectively. The laser power at the sample for NAD(P)H and FAD excitation was approximately 2.3 mW and 7.9 mW, respectively. Fluorescence lifetime decays with 256 time bins were acquired across $256 \times 256$ pixel images with a pixel dwell time of 4.8 μs and an integration period of 60 s. All samples were illuminated through a 40×/1.15 NA objective (Nikon). FLIM was performed on differentiation day 0 (immediately pre-treatment with CHIR99021, a Wnt signaling activator), day 1 (24 h post-treatment with CHIR99021), day 3 (immediately pre-treatment with IWP2, a Wnt signaling inhibitor), and day 5 (48 h post-treatment with IWP2). For NKX2.5$^{EGFP/+}$ hPSCs, day 8 NAD(P)H lifetime variables were also collected. Two-photon excitation of NKX2.5-EGFP was performed at 890 nm and emission was collected with a 550/100 nm bandpass filter. A 500LP dichroic mirror was used. For the 2DG experiment, H9 embryonic stem cells were imaged before and 2 h after 10 mM 2DG treatment, respectively. For the rotenone experiment, H9 embryonic stem cells were imaged before and 15 min after 10 μM rotenone treatment, respectively. The instrument response function was acquired from the second harmonic generated signal of urea crystals at 890 nm and was measured for each imaging session. A step-by-step protocol describing the imaging process can be found at Protocol Exchange (https://doi.org/10.21203/rs.3.pex-1571/v1).

**Image analysis.** Lifetime images of NAD(P)H and FAD were analyzed via SPCImage software (Becker & Hickl). Two-component decays were calculated by the following equation[22]: $I(t) = \alpha_1 e^{-t/t_1} + \alpha_2 e^{-t/t_2} + C$. Fluorescence intensity images were generated by integrating photon counts over the per-pixel fluorescence decays. For FLIM analysis of cells from differentiation day 0 to day 5, pixels were binned to 1 ($3 \times 3$ pixels) to achieve good statistics for fluorescence decay fitting. Similarly, for NKX2.5-EGFP NAD(P)H lifetimes on day 8, pixels were binned to 2 ($5 \times 5$ pixels). The per-pixel ratio of NAD(P)H fluorescence intensity to FAD intensity was calculated to determine optical redox ratio. ASC files of FLIM variables were exported from SPCImage, then converted into TIF with ImageJ, and these TIF were imported to CellProfiler. A customized CellProfiler pipeline was used to segment individual cell cytoplasms[54]. Cytoplasm masks were applied to all images to determine single-cell optical redox ratio and NAD(P)H and FAD fluorescence lifetime variables. Fluorescence lifetime variables consist of the mean

lifetime ($\tau_m = \alpha_1 \tau_1 + \alpha_2 \tau_2$), free- and protein-bound lifetime components ($\tau_1$ and $\tau_2$ for NAD(P)H, and $\tau_2$ and $\tau_1$ for FAD, respectively), and their fractional contributions ($\alpha_1$ and $\alpha_2$; where $\alpha_1 + \alpha_2 = 1$) for each individual cell cytoplasm. A total 13 OMI variables were analyzed for each cell cytoplasm: FAD intensity, FAD $\alpha_1$, FAD $\alpha_2$, FAD $\tau_1$, FAD $\tau_2$, and FAD $\tau_m$; NAD(P)H intensity, NAD(P)H $\alpha_1$, NAD(P)H $\alpha_2$, NAD(P)H $\tau_1$, NAD(P)H $\tau_2$, and NAD(P)H $\tau_m$; optical redox ratio $= \frac{\text{NAD(P)H intensity}}{\text{FAD intensity}}$. The optical redox ratio is the relative fluorescence intensities of NAD(P)H and FAD and provides an optical measurement of the redox state of the cell[55–57]. Different optical redox ratio definitions with either NAD(P)H or FAD in the numerator can be found in the literature[55–57]. Here, we use NAD(P)H/FAD as the optical redox ratio.

The phasor plot of lifetime decays for enhanced green fluorescent protein (EGFP) and NAD(P)H was performed as previously described[58]. Briefly, phasor lifetime plots are derived from a Fourier transformation of fluorescence lifetime decay curves by a custom algorithm. The fluorescence lifetime of each pixel in the image is presented in a 2D phasor plot with the unitless horizontal axis (G) and the vertical axis (S).

**UMAP clustering.** Clustering of cells across all days and differentiation efficiency conditions was represented using UMAP. UMAP dimensionality reduction was performed on all 13 OMI variables (optical redox ratio, NAD(P)H $\tau_m$, $\tau_1$, $\tau_2$, $\alpha_1$, $\alpha_2$, and intensity; FAD $\tau_m$, $\tau_1$, $\tau_2$, $\alpha_1$, $\alpha_2$, and intensity) for projection in 2D space. The following parameters were used for UMAP visualizations: "n_neighbors": 10; "min_dist": 0.3, "metric": cosine or euclidean, "n_components": 2.

**Classification methods.** Logistic regression classifiers were trained to distinguish cells at low (< 50% cTnT+ on day 12) and high (≥ 50% cTnT+ on day 12) differentiation efficiency 1 day post-treatment with CHIR99021. Consistent separation of day 1 UMAP clusters from all other days across differentiation conditions prompted classification of single-cell autofluorescence data from day 1 differentiation. All day 1 OMI data were partitioned into two datasets from two independent biological replicates (dataset 1 and dataset 2). Dataset 1 was randomly partitioned into training and validation datasets at the proportions of 80% and 20%, respectively ($n = 8974$ cells in the training set, $n = 2244$ cells in the validation set). Dataset 2 was used to independently evaluate trained classifiers for predicting known differentiation efficiencies ($n = 19,245$ cells). Chi-squared variable selection was used to evaluate classification accuracy as a function of the number of training variables. Variable weights for OMI variables were extracted to determine the contribution of each variable to the trained logistic regression model. Receiver operating characteristic (ROC) curves were generated to evaluate the logistic regression model performance on classification of test set data. Support vector machine and random forest classifiers were also trained to classify low and high differentiation efficiencies on day 1 to determine whether classification performance was dependent on the chosen model. Training and test set partitioning and variable selection methods for support vector machine and random forest classifiers were identical to those reported for the logistic regression model.

**Z-score hierarchical clustering.** Z-score of each OMI variable for each condition was calculated across all 15 conditions. Z-score $= \frac{\mu_{observed} - \mu_{row}}{\sigma_{row}}$, where $\mu_{observed}$ is the mean value of each variable for each condition; $\mu_{row}$ is the mean value of each variable for all 15 conditions together; and $\sigma_{row}$ is the standard deviation of each variable across all 15 conditions. Heatmaps of z-scores for all OMI variables were generated to visualize differences in each variable between low and high differentiation efficiency conditions at day 0 and day 1. Dendrograms show clustering based on the similarity of average Euclidean distances across all variable z-scores. Heatmaps and associated dendrograms were generated in R (heatmap.2, gplots package).

**Statistics.** Data for OMI variables are presented as mean with 95% CI. Data for flow cytometry are presented as mean ± SEM. Statistical significance was determined by Student's $T$-test (two-tailed) between two groups. Three or more groups were analyzed by one-way analysis of variance (ANOVA) followed by Tukey's post hoc tests. $P < 0.05$ was considered statistically significant and indicated in the figures.

**Reporting summary.** Further information on research design is available in the Nature Research Reporting Summary linked to this article.

## Data availability

The authors declare that all relevant data supporting the findings of this study are available within the article and its Supplementary Information files. Original single cell OMI data are provided with this paper. Source data are provided and all Excel files used to generate and support the results from the associated code have been deposited in the following repository: https://github.com/skalalab/cardiomyocyte_differentiation (https://doi.org/10.5281/zenodo.5046151). A step-by-step protocol describing the differentiation method can be found at Protocol Exchange (https://doi.org/10.21203/rs.3.pex-1571/v1). Source data are provided with this paper.

## Code availability

All the code for generating UMAP, z-score heatmap, and classification results have been deposited in the following repository: https://github.com/skalalab/cardiomyocyte_differentiation (https://doi.org/10.5281/zenodo.5046151).

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

## Acknowledgements
This work was supported by the Morgridge Institute for Research. Some of the imaging optimization experiments were supported by the NIH (U01 EY032333, R01 CA211082, R01 CA205101, R01 CA185747) and NSF Center for Cell manufacturing Technologies (EEC-1648035). We thank Matthew Stefely for the help on figure illustration. We thank Jens Eickhoff for the useful suggestions on statistics.

## Author contributions
The project was conceived by T.Q. and M.C.S. The experiments were designed by T.Q. and M.C.S., and were carried out by T.Q. Data analysis was performed by T.Q., T.M.H., A.R.H., K.S. and K.S. The manuscript was written by T.Q., T.M.H. and M.C.S.

## Competing interests
The authors declare no competing interests.
