## [Peer Review File · Nature Communications]

Reviewers' Comments:

Reviewer #1:

Remarks to the Author:

This manuscript by the Skala group presents a label free autofluorescence imaging method to characterize stem cell differentiation efficiency. The ability to identify efficiency at day 12 using label-free imaging on day 1 is very exciting and a unique contribution. However there are some technical questions that need to be addressed:

1. There are 13 variables (some not independent of each other) and 11 conditions producing a range of differentiation efficiencies that are groups into high and low categories. Therefore there is a strong potential for overfitting during classifier training (SVM, random forest, logistic regression). I understand that ~25,000 cells were analyzed and split into training and testing datasets. However, these cells are not all independent of each other and if the datasets were partitioned by randomly assigning cells to a set, there is a significant concern. Cells from the same dish or same field are likely to have similar measurements. In other words, if a "test" cell can have its 4 nearest neighbors in the training set, the validation is not truly independent. The biological replicates from independent wells should be partitioned into the two sets to ensure an accurate and robust validation in which cells from the same well are not in both training and testing sets.

2. Although this study focuses on discrimination using day 1 results, it seems the difference between day 0 and day 1 OMI measurements is a much better discriminator. These are label free and nondestructive methods, so could the difference in a well be computed and used to predict differentiation efficiency with much greater accuracy?

3. Many results focus on comparisons between the highest and lowest efficiencies. Are the OMI measurements and differences conserved across the full spectrum of efficiencies? If NAD(P)H lifetime correlates with % cTnT+, it is a much more interesting study. This also reduces any concern that the differences we are seeing could be related to some of the shared initial conditions of the 3 high-efficiency groups (low seeding density and low CHIR99021) rather than differentiation efficiency. Predicting an actual efficiency % is much more powerful for optimizing manufacturing than a binary high/low prediction.

4. Is it a concern that the similar differentiation efficiencies don't cluster together in the dendrogram in Figure 2b? The 53.2 group seems pretty different from the 51.8 and 65.5 groups. The possible reason for the discrepancy may be the different cell lines (iPSC vs ESC). It also may be a donor issue. These concerns would be alleviated if there were more than a single ESC and iPSC line investigated in this study. Addressing my comments from the point above could also help alleviate this concern that sensitivity could be related to the interaction of a couple experimental conditions or cell lines rather than differentiation efficiency.

5. Is it possible to show NAD(P)H Tm and EGFP images from the same field of view and scale in Figure 4? The two images look similar, but I don't believe they are the same field. It would be interesting to see how related the measurements are to each other on a cell-to-cell basis.

6. Similarly, it is interesting that there are strong differences on Day 8 (larger than on Day 1 as shown in Figure 4), but this was not the case in Fig. S6. What could account for this difference in results between experiments?

7. Figure S7 provides compelling evidence that you can separate EGRP pixels that have a mono-exponential decay from the bi-exponential decay of NADPH. However I don't understand how EGRP imaging was performed. What wavelengths were used? If it was collected from the same channel as NADPH, why does the phasor diagram indicate the two fluorophores do not spatially overlap?

8. What dichroics were used in the microscope?

9. I did not follow the argument in the discussion that these results confirm OMI methods can distinguish CMs and non-CM cells. Could the authors demonstrate an ability to distinguish CMs and fibroblasts in their images?

10. The authors indicate these changes in NADH and FAD lifetime may be related to glycolysis. However, it is not clear how FAD lifetime changes in response to 2DG, and it is not shown how either set of lifetime parameters change in response to rotenone. A number of standard assays could also confirm whether glycolysis is increasing in these differentiation conditions at Day 1. Some comparison of lifetime to metabolic assays would help support the statements connecting Wnt activation, glycolysis, and autofluorescence imaging.

11. Various groups have used OMI variables to assess stem cell differentiation. It would help to place this current work in the context of other studies. Is this potentially applicable to differentiation along other lineages? Broadly speaking, the appeal of the current study seems to be the sensitivity on day 1, which perhaps no group has shown before with any differentiation protocol.

Reviewer #2:

Remarks to the Author:

The manuscript «Label-free imaging for quality control of cardiomyocyte differentiation» aims to estimate non-invasive label-free method to predict the outcome of human pluripotent stem cells (hPSC) differentiation into cardiomyocytes (CM) based on live cell autofluorescence lifetime imaging, single-cell image analysis, and machine learning. The study presents interesting data about the high prediction accuracy and the model performance with 13 optical metabolic imaging variables combined across 11 different differentiation conditions including 3 hPSC lines. Live cell autofluorescence imaging and multivariate classification models provided high accuracy to separate low (<50%) and high ($\geq 50\%$) differentiation efficiency groups (quantified by cTnT expression on day 12) within 1 day after initiating differentiation. These results revealed that autofluorescence imaging can separate CM differentiation efficiencies at an early stage based on the metabolic changes. Your work is useful and very important for the development of new approaches for the non-invasive monitoring of hPSC differentiation. I recommend this for publication in Nature Communications pending the address of some major concerns.

Major concern:

- 1) Were pixels binned for analysis? Was an instrument response function of the system acquired and used in the lifetime analysis? It would be more representative if authors add this information in Materials and Methods (Image analysis).
- 2) Initial studies use $(FAD/NAD(P)H + FAD)$ to calculate the redox ratio because it normalized variations in the data (Georgakoudi I., 2012). Authors may consider justifying their use of the older method $(NAD(P)H/FAD)$ for this study.
- 3) The originality of the approaches to clustering and machine-learning algorithm based on metabolic parameters used by the authors is not completely clear. So, in the work (Hao Zhou et al. "Non-invasive Optical Biomarkers Distinguish and Track the Metabolic Status of Single Hematopoietic Stem Cells", Cell Press, Volume 23, Issue 2, 2020) were used the same approaches for the metabolism-based sorting of stem cells during differentiation. It would be helpful if the authors can include in the discussion more information about FLIM metabolic imaging in stem cell research and approaches to segmentation and machine-learning algorithm used by other researchers for stem cell metabolic sorting.
- 4) Could this methodology be adapted potentially for in vivo assessment of the early stage of stem cell differentiation using intravital imaging?

Reviewer #3:

Remarks to the Author:

The manuscript by Qian et al. describes an elegant biotechnology improvement for the in-vitro differentiation of human pluripotent cells into cardiomyocytes.

The investigators only vary the conditions at the beginning of the protocol (seeding density and concentration of Wnt pathway agonist). It is therefore not surprising that the authors find that varying those early conditions in the protocol will influence the outcome of the differentiation, but

many other things may as well, or possibly even more so, which is not explored in this manuscript.

However, this investigation is really designed to investigate how early during the differentiation protocol one can measure whether the early conditions are suitable for subsequent cardiomyocyte differentiation. While not introducing this research question at all well in the manuscript, the authors results are really spectacular: They can tell already after 24 hours (from metabolic information collected by measuring and imaging autofluorescence from many NAD(P)H and FAD molecules), and they can do this at a single-cell level. The discovery that successfully treated cells can be recognised in a recognisable Day1 cluster is unfortunately somehow buried in the supplemental information (currently Figure S3b).

While further discrimination between the 13 measurable variables in their data, the authors highlight that the NAD(P)H tau m variable alone is already somewhat accurate at predicting subsequent differentiation success (e.g. Fig.3c); however, what is for practical applications probably more important is that measuring just two variables NAD(P)H tau m and FAD tau m is pretty much as accurate as measuring all 13 (which is also consistent with what the authors then use in subsequent experiments, e.g. Fig.4).

Minor Points:

- Consider providing a visual outline of the established differentiation protocol, highlighting the steps varied in this investigation and introducing this investigation's focus on just studying varying early conditions in this protocol.
- Consider introducing this study's research question on how early the influence of these early conditions on subsequent differentiation success can be monitored
- Consider whether what is currently Fig S3 should be presented in the main manuscript to document the discovery of a reliable signal at Day1 already.
- Consider whether more cell lines should be tested.
- Consider whether the clustering in Fig.2b is influenced by the nature of the cell line (H9 ESC vs IMR90 iPSC) as much as by the differentiation efficiency
- Consider whether abbreviations such as ROC and AUC should again be introduced in the result section (these are introduced in Introduction and Figure legend).
- Consider whether panel lettering in current Figs 1 and 4 should be more comparable.
- Consider whether current Figure 4 (experiments with NKX2.5EGFP) could be presented as suppl. material, or whether focus on measuring metabolic variables at Day8 should be better introduced and relationship with cardiac-like fibroblast differentiation better explored and possibly documented within these results.
- Consider whether investigating how CHIR99021 influences the altered metabolic indicators could be relevant (e.g. is the altered metabolism directly caused by Wnt signalling (in parallel with mesoderm induction) or downstream of successful mesoderm induction?)

**Responses to Review of Nature Communications manuscript
NCOMMS-20-44445 by Qian et al.**

We appreciate the reviewers' thoughtful comments and specific suggestions to improve our manuscript. To address these comments, we have performed additional experiments and analysis and carefully revised the manuscript text as described in detail below.

Specifically, to address the comments about adding more cell lines from Reviewer 1 and Reviewer 3, we performed four more differentiation conditions with two additional hPSC lines. Regarding Reviewer 1's comment about the overfitting issue, we followed the reviewer's suggestion and partitioned the data into two independent datasets for classifier training and prediction. Below, comments from the reviewers and editor are shown in ***bold italic font***. Our corresponding responses are shown in black font. Changes in the manuscript file are in red font and new figure panels and edited manuscript text are directly pasted in the response document to facilitate their evaluation. Additionally, blue font indicates changes made in the manuscript for clarity.

Thank you again for submitting your manuscript "Label-free imaging for quality control of cardiomyocyte differentiation" to Nature Communications. We have now received reports from 3 reviewers and, on the basis of their comments, we have decided to invite a revision of your work for further consideration in our journal. Your revision should address all the points raised by our reviewers (see their reports below). Specifically, as noted by Reviewer #1, you must show that your results are independently validated and robustly sensitive to differentiation efficiency. We also ask that you add some insight on the biology behind these metabolic changes during differentiation, as requested by Reviewers #1 and #3, and add additional cell lines to your analysis.

When resubmitting, you must provide a point-by-point response to all of the reviewers' comments. Please show all changes in the manuscript text file with track changes or colour highlighting. If you are unable to address specific reviewer requests or find any points invalid, please explain why in the point-by-point response.

Important: In addition to the above, you must comply with the following editorial requests; we will not be able to proceed with your revised manuscript otherwise. Please also see the Nature Communications formatting instructions, which you may find useful while preparing your revised manuscript.

*** Please replace your bar graphs with plots that feature information about the distribution of the underlying data. All data points should be shown for plots with a sample size less than 10. For larger sample sizes, please consider box-and-whisker or violin plots as alternatives. Measures of centrality, dispersion and/or error bars should be plotted and described in the figure legend.**

We thank the editor for the opportunity to resubmit our manuscript to *Nature communications* for consideration as a research article. We have addressed the reviewers' comments point by point. Specifically, we included the data from two additional human pluripotent stem cell lines and four more conditions to address the reviewers' comments about adding more cell lines to validate the method. We also performed optical metabolic imaging data classification by partitioning the data into two independent biological replicates following the reviewer's suggestion to show that our results are independently validated and robustly sensitive to differentiation efficiency. With this new data and classification method, we achieve an accuracy score over 85% with an AUC of ROC above 0.90 for binary prediction of high ($\geq 50\%$) vs. low ($< 50\%$) cardiomyocyte differentiation efficiency from human pluripotent stem cells. Note that the additional experiments were performed over two years apart from the original experiments. Considering the changes with instruments and laser settings during these two years, this consistent trend in high-performance classification supports the utility of this method for distinguishing high and low differentiation efficiency from multivariate label-free optical metabolic imaging data. We have also included new data on metabolism during the first 24 hours of differentiation, using standard measurements of glucose consumption and lactate secretion (Fig. 1) to provide insight into the biology behind these metabolic changes during differentiation.

We also followed the journal's requirement and replaced all the bar graphs with dot plots.

All of the responses and changes are included in greater detail below.

REVIEWER COMMENTS

Reviewer #1 (Remarks to the Author):

This manuscript by the Skala group presents a label free autofluorescence imaging method to characterize stem cell differentiation efficiency. The ability to identify efficiency at day 12 using label-free imaging on day 1 is very exciting and a unique contribution. However there are some technical questions that need to be addressed:

1. There are 13 variables (some not independent of each other) and 11 conditions producing a range of differentiation efficiencies that are grouped into high and low categories. Therefore, there is a strong potential for overfitting during classifier training (SVM, random forest, logistic regression). I understand that ~25,000 cells were analyzed and split into training and testing datasets. However, these cells are not all independent of each other and if the datasets were partitioned by randomly assigning cells to a set, there is a significant concern. Cells from the same dish or same field are likely to have similar measurements. In other words, if a "test" cell can have its 4 nearest neighbors in the training set, the validation is not truly independent. The biological replicates from independent wells should be

partitioned into the two sets to ensure an accurate and robust validation in which cells from the same well are not in both training and testing sets.

We appreciate the reviewer's concern that the validation might not be truly independent. Hence, we followed the reviewer's suggestion and partitioned single cell OMI data from differentiation day 1 into two independent datasets. Cells from biological replicate 1 (dish 1) were assigned to dataset 1, and cells from biological replicates 2&3 were assigned to dataset 2. Together with two additional hPSC lines and four more conditions (15 conditions in total), 80% of cells from dataset 1 (8974 cells) were used for training, and 20% of cells from dataset 1 (2244 cells) were used as the validation set to determine classifier performance during hyperparameter tuning. All cells from dataset 2 (19245 cells) were used for evaluation of trained classifiers for predicting known differentiation efficiencies. With this additional data and dataset partitioning, we were still able to achieve area under the receiver operating characteristic curve above 0.90 with all three classifiers (logistic regression, SVM, and random forest). We edited the main text with updated results and classification methods and updated Table 1, Figure 2, Figure 3, Figure S3, and Figure S4 accordingly.

FROM RESULTS: (P8) "Next, day 1 data were separated into two datasets for classification training (dataset 1), validation (dataset 1), and prediction (dataset 2), where dataset 1 and 2 are independent biological replicates. A logistic regression classifier based on all 13 OMI variables was trained on 80% of the single cell data from dataset 1 to classify cells from low vs. high differentiation efficiency conditions and validated on the remaining 20% of single cell data from dataset 1. Reported variable weights and accuracy scores were validated from dataset 1. Performance of classifiers (receiver operating characteristic (ROC) area under the curve (AUC)) was evaluated on dataset 2. Variable weights indicated that NAD(P)H τ_m and FAD τ_m were important variables for discriminating low vs. high differentiation efficiency conditions (Figure 3b). Logistic regression (Figure 3c, d), support vector machine (Figure S4a, b), and random forest (Figure S4c, d) classifiers were generated to test the prediction accuracy using all 13 OMI variables, yielding an accuracy score > 85% for all three classifiers. ROCs based on logistic regression classifiers using all 13 OMI variables and a subset of variables are shown in Figure 3e along with their performance defined by the AUC. Here, the logistic regression classifier using all 13 OMI variables achieves an AUC > 0.90 (Figure 3e). With only NAD(P)H lifetime variables (NAD(P)H τ_m , τ_2 , α_2 , α_1 , τ_1) that can be collected in the NAD(P)H channel alone, the AUC is > 0.83 (Figure 3e)."

FROM MATERIALS AND METHODS: (P18) "All day 1 OMI data were partitioned into two datasets from two independent biological replicates (dataset 1 and dataset 2). Dataset 1 was randomly partitioned into training and validation datasets at proportions of 80% and 20%, respectively (n = 8974 cells in the training set, n = 2244 cells in the validation set). Dataset 2 was used to independently evaluate trained classifiers for predicting known differentiation efficiencies (n = 19245 cells)."

Figure 3. OMI variables accurately distinguish cells under low or high cardiomyocyte differentiation efficiency conditions on day 1. (a) UMAP dimensionality reduction was performed on all 13 OMI variables (optical redox ratio, NAD(P)H τ_m , τ_1 , τ_2 , α_1 , α_2 , and intensity; FAD τ_m , τ_1 , τ_2 , α_1 , α_2 , and intensity) for each cell on day 1 and projected onto 2D space. Day 1 cells from all 15 conditions shown in Table 1 are plotted together with cells from low (< 50% cTnT+ on day 12) and high (\geq 50% cTnT+ on day 12) CM differentiation efficiencies in light gray and dark gray, respectively. $n = 16048$ and 14415 cells for low and high differentiation efficiency conditions, respectively. (b-e) All OMI data from day 1 cells were separated into two datasets. Dataset 1 was randomly partitioned into 80% portions for training and 20% portions for testing, respectively ($n = 8974$ cells for training, $n = 2244$ cells for validation). Dataset 2 was used for evaluation of classifier performance. Binary classification was tested for low (< 50% cTnT+) vs. high (\geq 50% cTnT+) differentiation efficiency conditions on day 1. (b) OMI variable weights are shown specific to the logistic regression model. (c) Classification accuracy with respect to number of OMI variables was evaluated by chi-squared variable selection to separate low and high differentiation efficiency conditions with the logistic regression model. The number of variables included in the logistic regression model are indicated at bottom-axis. Data are presented as mean \pm STDEV. (d) The variables included for each logistic regression model [specified by numbers of variables on the x-axis in (c)] are defined, where the blue text indicates NAD(P)H lifetime variables and the red text indicates FAD lifetime variables. The OMI variables included in each instance (e.g., 3, 4) are indicated by a light blue + in each column. (e) Model performance of the logistic regression classifier was evaluated by receiver operating characteristic (ROC) curves using different OMI variable combinations as labelled. The area under the curve (AUC) is provided for each variable combination as indicated in the legend.

2. Although this study focuses on discrimination using day 1 results, it seems the difference between day 0 and day 1 OMI measurements is a much better discriminator. These are label free and nondestructive methods, so could the difference in a well be computed and used to predict differentiation efficiency with much greater accuracy?

We appreciate the reviewer's suggestion that the difference between day 0 and day 1 measurements might be a better discriminator. Cells undergoing differentiation from day 0 to day 1 were migrating and dividing. Therefore, we were not able to longitudinally track the change of each individual cell. Hence, we normalized all OMI data from day 1 to corresponding mean OMI values from day 0 for each condition to perform UMAP, data classification, and ROC curve evaluation. We evaluated this day 0 normalization approach using the same data partitioning as described above. Briefly, cells from biological replicate 1 were assigned to dataset 1, and cells from biological replicates 2&3 were assigned to dataset 2. UMAP clustering and AUC values of ROC curves for day 0 normalized data are slightly inferior to that with non-normalized data (FIG. A and FIG. B, below). This outcome is likely because we were not able to track the changes of each individual cell due to cell migration and high proliferation of hPSCs. Human pluripotent stem cells exhibit high proliferation rate and short cell cycle of approximately 16 hours (*Nat Cell Biol.* 2019, 21(9): 1060–1067.; *Curr Biol.* 2011, 21(1): 45–52.). Cell migration contributes to the colony formation efficiency (*Regen. Ther.* 2019, 10: 27-35). Newly divided cells might contribute to the separated clustering for low and high differentiation efficiency conditions. Hence, day 1 alone would be more efficient to predict the low and high differentiation efficiency outcome.

FIG. A. Normalized OMI variables from day 1 to day 0 poorly separate cells under low or high cardiomyocyte differentiation efficiency conditions. Day 1 OMI variables were normalized to mean values of day 0 OMI variables for the same condition. UMAP dimensionality reduction was performed on all 13 normalized day 1 OMI variables (optical redox ratio, NAD(P)H τ_m , τ_1 , τ_2 , α_1 , α_2 , and intensity; FAD τ_m , τ_1 , τ_2 , α_1 , α_2 , and intensity) for each cell and projected onto 2D space. Day 1 cells from all 15 conditions shown in Table 1 are plotted together with cells from low ($< 50\%$ cTnT+ on day 12) and high ($\geq 50\%$ cTnT+ on day 12) CM differentiation efficiencies in blue and red, respectively.

3. Many results focus on comparisons between the highest and lowest efficiencies. Are the OMI measurements and differences conserved across the full spectrum of efficiencies? If NAD(P)H lifetime correlates with % cTnT+, it is a much more interesting study. This also reduces any concern that the differences we are seeing could be related to some of the shared initial conditions of the 3 high-efficiency groups (low seeding density and low CHIR99021) rather than differentiation efficiency. Predicting an actual efficiency % is much more powerful for optimizing manufacturing than a binary high/low prediction.

We appreciate the reviewer's comments. We then carefully checked the OMI difference between the low and high differentiation efficiency conditions. The day 1 OMI variable z-score heatmap indicates that OMI differences are conserved in a majority of high and low differentiation efficiency conditions, but not perfectly conserved across the full spectrum, such as new condition at 61.0% differentiation efficiency (Figure 2b). The OMI variable differences could be caused by multiple factors, including seeding density, CHIR99021 concentration, and behavior from different cell lines. Multi-dimensional data projection onto 2D and classification models performed well for binary prediction of high and low differentiation efficiencies. Therefore, we used multivariate OMI variable classification to predict differentiation outcome.

We agree with the reviewer's suggestion that prediction of actual differentiation efficiency would be more powerful and useful for cell manufacturing. Hence, we attempted 2 different methods for predicting more defined classes of differentiation efficiency. First, we trained multiclass classifiers (logistic regression, SVM, random forest) with the differentiation efficiencies separated into four classes (0%-20%, 20-40%, 40-60%, 60%-80%) and evaluated their classification accuracies. Second, we performed principal component analysis on all 13 OMI variables for dimensionality reduction and regressed on the resultant PCA scores from each of the first 3 principal components to assess correlation between the composite multivariate OMI data and differentiation efficiency. Classification using multiclass classifiers do not perform well, as shown from low classification accuracies reported in FIG. C, below. Also, PCA regression analysis showed poor correlation of all principal components with differentiation efficiencies. The morphology of different cell lineages differentiated from hPSCs differ (*Nature Methods*, 2015, 12: 637–640). Hence, variables like morphology could be introduced to establish a more robust prediction model with high accuracy in the future. We modified the main text:

FROM RESULTS: (P7) "Heatmap dendrogram clustering based on OMI variable z-scores revealed that cells under high differentiation efficiency conditions on day 1 were clustered closely together and distinct from cells under low differentiation efficiency conditions on day 1 (Figure 2b) except the 61.0% differentiation efficiency with 6 μ M CHIR99021 treatment for 19-9-11 hPSCs. This outlier could be due to multiple reasons, e.g., the low CHIR99021 concentration or low cell seeding density. In the future, more variables, such as cell morphology, can be introduced to increase linear clustering performance."

FIG. C. PCA analysis and classification models do not perform well to predict intervals of differentiation efficiency. Principal component analysis was performed for dimensionality reduction of all 13 OMI variables prior to performing linear regression on PCA scores from each of the first 3 principal components with respect to differentiation efficiency percentages. These 3 components were selected to preserve the greatest variance of the original data. The red lines represent each linear regression model fit. Shaded regions represent the 95% confidence intervals of the fit. R-squared values indicated goodness of model fit. Multiclass logistic regression, SVM, and random forest classifiers were trained using all 13 OMI variables or a subset of variables (indicated in table) for distinguishing 4 classes of differentiation efficiency ranges (0%-20%, 20-40%, 40-60%, 60%-80%). Classification accuracies for each multiclass classifier represented respective classifier performance.

4. Is it a concern that the similar differentiation efficiencies don't cluster together in the dendrogram in Figure 2b? The 53.2 group seems pretty different from the 51.8 and 65.5 groups. The possible reason for the discrepancy may be the different cell lines (iPSC vs ESC). It also may be a donor issue. These concerns would be alleviated if there were more than a single ESC and iPSC line investigated in this study. Addressing my comments from the point above could also help alleviate this concern that sensitivity could be related to the interaction of a couple experimental conditions or cell lines rather than differentiation efficiency.

We agree with the reviewer's concern and appreciate the reviewer's suggestion. Hence, we performed differentiation with two more hPSC lines and four more conditions. The z-

score heatmap clustered well with the exception of the new 61.0% differentiation efficiency condition. Heatmap clustering was performed based on similarity of average Euclidean distances across all variable z-scores for 15 differentiation conditions. The exception of z-score clustering possibly is due to different reasons, such as z-score clustering is a linear regression, and new conditions were optimized with much lower CHIR99021 concentrations and different seeding density. In the Z-score heatmap clustering, NAD(P)H τ_m is conserved very well through the full spectrum of differentiation efficiencies (Figure 2b) and it is consistent with the logistic regression classifier (Figure 3). However, for the purpose of high prediction accuracy, more OMI variables are needed. In multivariate clustering, UMAP and logistic regression classifier performed better than z-score heatmap. For new hPSC lines, different seeding densities and CHIR99021 concentrations were optimized for differentiation, which is consistent with previous studies that showed that different conditions are needed to optimize cardiomyocyte differentiation for different cells of origin (*Cell stem cell*, 2018, 23(4): 586-598; *Proc. Natl. Acad. Sci.* 2012, 109: E1848-E1857). Note that the additional experiments were performed over two years apart from the original experiments. During these two years, instrument settings and laser features have been changed, so changes in equipment could also be a factor affecting the z-score heatmap. However, we were able to achieve high prediction accuracies with different classification models trained on the multivariate data, yielding an AUC of ROC above 0.9. Considering the changes with instruments and laser setting during these two years, this consistent trend in high-performance classification supports the utility of this method for distinguishing high and low differentiation efficiency from multivariate OMI data. We updated the manuscript includes this new data.

ROM RESULTS: (P7) "Heatmap dendrogram clustering based on OMI variable z-scores revealed that cells under high differentiation efficiency conditions on day 1 were clustered closely together and distinct from cells under low differentiation efficiency conditions on day 1 (Figure 2b) except the 61.0% differentiation efficiency with 6 μ M CHIR99021 treatment for 19-9-11 hPSCs. This outlier could be due to multiple reasons, e.g., the low CHIR99021 concentration or low cell seeding density. In the future, more variables, such as cell morphology, can be introduced to increase linear clustering performance."

Figure 2. Multivariate analysis reveals unique metabolic profiles in cells differentiated into cardiomyocytes at day 1. (a) Uniform Manifold

Approximation and Projection (UMAP) dimensionality reduction was performed on all 13 autofluorescence variables (optical redox ratio, NAD(P)H τ_m , τ_1 , τ_2 , α_1 , α_2 , and intensity; FAD τ_m , τ_1 , τ_2 , α_1 , α_2 , and intensity) for each cell and projected onto 2D space. Cells from all 11 conditions shown in Table 1 are plotted together. Data include cells from day 0, day 1, day 3, and day 5. Each dot represents one single cell, and $n = 25304, 25470, 26228,$ and 23484 cells for day 0, 1, 3, and 5, respectively. (b) Heatmap dendrogram clustering based on similarity of average Euclidean distances across all variable z-scores was performed on day 1 cells across all 15 conditions. Conditions are indicated by the CM differentiation efficiency percentages as noted by column labels at the top of the heatmap (quantified by flow cytometry cTnT+ on day 12, full conditions given in Table 1). Low differentiation efficiencies ($< 50\%$) are shaded in light gray and high differentiation efficiencies are shaded in dark gray ($\geq 50\%$). Z-score = $\frac{\mu_{observed} - \mu_{row}}{\sigma_{row}}$, where $\mu_{observed}$ is the mean value of each variable for each condition;

μ_{row} is the mean value of each variable for all 15 conditions together, and σ_{row} is the standard deviation of each variable across all 15 conditions. Autofluorescence variables are indicated on the left side as row labels. $n = 30463$ cells from day 1. (c) Separated UMAP clusters for representative differentiation conditions. Conditions are labelled with original cell seeding density, CHIR99021 treatment concentration, and final cardiomyocyte differentiation efficiency quantified by flow cytometry (detailed in Table 1). $n = 13897, 13852, 4357,$ and 7601 cells for condition 65.5%, 51.8%, 19.6%, 15.1%, respectively.

5. Is it possible to show NAD(P)H Tm and EGFP images from the same field of view and scale in Figure 4? The two images look similar, but I don't believe they are the same field. It would be interesting to see how related the measurements are to each other on a cell-to-cell basis.

We appreciate the reviewer's comments. The images in both Figure 4i and Figure 4j are from the same field of view. Due to the different localizations of NAD(P)H and NKX2.5-EGFP, it seems that these two images were not from the same field of view. In order not to confuse the audiences, we merged images and updated Figure 4.

Figure 4

6. Similarly, it is interesting that there are strong differences on Day 8 (larger than on Day 1 as shown in Figure 4), but this was not the case in Fig. S6. What could account for this difference in results between experiments?

We agree with the reviewer's comment that the OMI difference in cells on day 8 is larger than the difference in cells on day 1. Cells on day 1 are still at very early stage of differentiation. After CHIR99021 treatment for 24 hours, all the cells are differentiated into primitive streak, but these early progenitor cells have different potentials to differentiate into different cell lineages based on the Wnt activation status by CHIR99021. Cells on day 8 are at the end of differentiation stage into distinct populations, including cardiomyocytes and possibly fibroblasts. Hence, OMI differences are larger on day 8. We added more discussion about this difference in the discussion.

FROM DISCUSSION: (P12) "OMI distinguished different cell populations at multiple time-points in this differentiation protocol, with larger differences between low and high differentiation efficiencies on day 8 than on day 1. Day 1 cells are mainly primitive streak whereas cells on day 8 are at the end of differentiation and exhibit the glycolytic activities of fetal/newborn CMs⁴⁵. Therefore, larger differences are expected in OMI between CMs and non-CMs at day 8 compared to low and high differentiation efficiencies on day 1. OMI differences between low and high differentiation efficiencies at multiple days in this protocol indicates that this technology could continuously monitor stem cell differentiation stages."

7. Figure S7 provides compelling evidence that you can separate EGRP pixels that have a mono-exponential decay from the bi-exponential decay of NADPH. However I don't understand how EGRP imaging was performed. What wavelengths were used? If it was collected from the same channel as NADPH, why does the phasor diagram indicate the two fluorophores do not spatially overlap?

We thank the reviewer for the comment about how we imaged EGFP. Two-photon excitation of EGFP was performed at 890 nm and signal was collected at 550 ± 50 nm. Two-photon excitation of NAD(P)H was performed at 750 nm and signal was collected at 440 ± 40 nm. Figure S7 (currently Figure S6) showed that the NAD(P)H fluorescence at 750nm excitation is free from EGFP fluorescence signal. We added a reference to support that our measured EGFP lifetime is consistent with previously published values.

FROM RESULTS: (P10) "After confirming that EGFP lifetimes are consistent with previous measurements³⁸ and do not overlap with NAD(P)H lifetimes (Figure S6), we then evaluated the NAD(P)H lifetimes of differentiated CMs on day 8 when the cells expressed NKX2.5-EGFP."

8. What dichroics were used in the microscope?

We thank the reviewer for this question. The dichroic mirror we used in the experiment is 500LP. We added the information in the methods.

FROM MATERIALS AND METHODS: (P16) “Two-photon excitation of NKX2.5-EGFP was performed at 890 nm and emission was collected with a 550/100 nm bandpass filter. A 500LP dichroic mirror was used.”

9. I did not follow the argument in the discussion that these results confirm OMI methods can distinguish CMs and non-CM cells. Could the authors demonstrate an ability to distinguish CMs and fibroblasts in their images?

We thank the reviewer for these comments. The statement in the discussion that OMI can distinguish CMs and non-CMs is based on the result showed in Figure 4h-j with the NKX2.5-EGFP reporter line. Cells that differentiate into CMs express NKX2.5-EGFP, and non-CMs do not express EGFP. Therefore, the statement about OMI distinguishing CMs and non-CMs is true for this reporter line, but not necessarily true in primary cells. Hence, we modified the text in the discussion.

FROM DISCUSSION: (P12) “These results further confirm that autofluorescence imaging can distinguish the distinct metabolic activities between hPSC-derived CMs and other non-CMs with the NKX2.5-EGFP reporter line.”

10. The authors indicate these changes in NADH and FAD lifetime may be related to glycolysis. However, it is not clear how FAD lifetime changes in response to 2DG, and it is not shown how either set of lifetime parameters change in response to rotenone. A number of standard assays could also confirm whether glycolysis is increasing in these differentiation conditions at Day 1. Some comparison of lifetime to metabolic assays would help support the statements connecting Wnt activation, glycolysis, and autofluorescence imaging.

We thank the reviewer for the comment about the relationship between OMI lifetimes and glycolysis. In order to confirm the connection of Wnt activation, glycolysis, and OMI variables, glycolysis and lactate assays were performed for cell culture medium from low and high differentiation efficiency conditions. Glycolysis assay and lactate assay results showed that high differentiation efficiency conditions had more glycolytic activity. We updated Figure 1 and updated text in the results.

RESULTS: (P6) “Lactate and glucose concentrations in the cell culture medium after 24 hours of differentiation for low and high differentiation efficiency conditions revealed more glycolytic activity in the high differentiation efficiency condition (Figure 1n, o). Changes in autofluorescence with known metabolic inhibitors and during CM differentiation together with lactate and glucose assay results indicate that differentiating cells altered their metabolic activity 1 day after CHIR99021 treatment.”

Figure 1. See captions on the next page.

Figure 1. NAD(P)H and FAD fluorescence change differently and reflect different glycolytic activities in the first 24 hours for cells in low vs. high cardiomyocyte differentiation efficiency conditions. hPSCs were differentiated into CMs following an established method¹¹. On differentiation day 12, cells were verified by flow cytometry with cTnT labelling from three independent replicates. **(a, b)** Representative flow cytometry dot plots for **(a)** low, and **(b)** high differentiation efficiencies along with negative controls. Single-cell quantitative analysis of mean lifetimes (τ_m , reported as picoseconds) of **(c, d, e)** FAD and **(f, g, h)** NAD(P)H, and **(i, j, k)** optical redox ratio for low (0.3% cTnT+) and high differentiation (65.5% cTnT+) efficiencies on day 0 (“D0”, immediately pre-treatment) and day 1 (“D1”, 24 hours post-treatment with CHIR99021), and their corresponding representative images. $n = 2458, 633, 3534$ and 4446 cells for 0.3% day 0, 0.3% day 1, 65.5% day 0, and 65.5% day 1, respectively. Data are presented as dot plots with bars for the mean and 95% CI for each condition each day. Statistical significance was determined by one-way analysis of variance (ANOVA) followed by Tukey’s post hoc tests. **** $p < 0.0001$. Color bars are indicated on the right. Changes of optical redox ratio after treatment with 2DG or rotenone. **(l)** Single-cell quantitative analysis of optical redox ratio for H9 ESCs before and 2 hours after 10 mM 2DG treatment. $n = 1051$ and 900 cells for before and after 2DG treatment, respectively. **(m)** Single-cell quantitative analysis of optical redox ratio for H9 ESCs before and 15 minutes after 10 μ M rotenone treatment. $n = 1042$ and 986 cells for before and after rotenone treatment, respectively. Data are presented as dot plots with bars for the mean and 95% CI. Statistical significance was determined by Student’s T-test. **** $p < 0.0001$. ps, picoseconds. After the first 24 hours of differentiation, **(n)** lactate and **(o)** glucose concentrations of cell culture medium from low (10.8%) and high (63.1%) differentiation efficiency conditions were measured with three biological replicates, respectively. Data are presented as dot plots with mean \pm SEM. Statistical significance was determined by Student’s T-test. * $p < 0.05$.

11. Various groups have used OMI variables to assess stem cell differentiation. It would help to place this current work in the context of other studies. Is this potentially applicable to differentiation along other lineages? Broadly speaking, the appeal of the current study seems to be the sensitivity on day 1, which perhaps no group has shown before with any differentiation protocol.

We appreciate the reviewer’s comments about relating our study to other stem cell differentiation studies using autofluorescence. Hence, we added more of these important references to the introduction as they established the feasibility of the current study. Most of these previous studies monitored autofluorescence changes between different cell lineages during stem cell differentiation. Our study is distinct from these previous studies because we build a predictive model based on OMI that can determine

low (<50%) and high (≥50%) differentiation efficiencies at day 1. We have also clarified this point in the discussion.

FROM INTRODUCTION: (P3) “Several groups have demonstrated that autofluorescence lifetime imaging can non-invasively track stem cell metabolic activities in real time, including monitoring mesenchymal stem cell differentiation into adipocytes^{23, 24}, osteocytes^{24, 25}, and chondrocytes²⁵, distinguishing differentiation of hPSCs into dermal and epidermal lineages²⁶, metabolic difference between hPSCs and feeder cells²⁷, and hematopoietic stem cells at different stages²⁸. These prior studies indicate that OMI is suitable to detect the metabolic changes that occur during CM differentiation.”

FROM INTRODUCTION: (P4) “Compared to previous studies²³⁻²⁸, we specifically contribute a predictive model based on OMI to determine CM differentiation outcome as early as day 1.”

Reviewer #2 (Remarks to the Author):

The manuscript «Label-free imaging for quality control of cardiomyocyte differentiation» aims to estimate non-invasive label-free method to predict the outcome of human pluripotent stem cells (hPSC) differentiation into cardiomyocytes (CM) based on live cell autofluorescence lifetime imaging, single-cell image analysis, and machine learning. The study presents interesting data about the high prediction accuracy and the model performance with 13 optical metabolic imaging variables combined across 11 different differentiation conditions including 3 hPSC lines. Live cell autofluorescence imaging and multivariate classification models provided high accuracy to separate low (<50%) and high (≥ 50%) differentiation efficiency groups (quantified by cTnT expression on day 12) within 1 day after initiating differentiation. These results revealed that autofluorescence imaging can separate CM differentiation efficiencies at an early stage based on the metabolic changes. Your work is useful and very important for the development of new approaches for the non-invasive monitoring of hPSC differentiation. I recommend this for publication in Nature Communications pending the address of some major concerns.

Major concern:

1) Were pixels binned for analysis? Was an instrument response function of the system acquired and used in the lifetime analysis? It would be more representative if authors add this information in Materials and Methods (Image analysis).

We appreciate the reviewer's comment about the image analysis. For image analysis from day 0 to day 5, pixels were binned at 1 (3 × 3 pixels). For NKX2.5-EGFP line day 8 image analysis, pixels were binned at 2 (5 × 5 pixels). The instrument response function was acquired for each imaging session. We have added these details in the Materials and Methods.

FROM MATERIALS AND METHODS: (P16) “The instrument response function was acquired from the second harmonic generated signal of urea crystals at 890 nm and was measured for each imaging session.”

FROM MATERIALS AND METHODS: (P16) “For FLIM analysis of cells from differentiation day 0 to day 5, pixels were binned to 1 (3 × 3 pixels) to achieve good statistics for fluorescence decay fitting. Similarly, for NKX2.5-EGFP NAD(P)H lifetimes on day 8, pixels were binned to 2 (5 × 5 pixels).”

2) Initial studies use (FAD/NAD(P)H + FAD) to calculate the redox ratio because it normalized variations in the data (Georgakoudi I., 2012). Authors may consider justifying their use of the older method (NAD(P)H/FAD) for this study.

We thank the reviewer’s comment about the calculation of redox ratio. The optical redox ratio is the relative fluorescence intensities of NAD(P)H and FAD, and provides an optical measurement of the redox state of the cell (*Chance 1979, Varone 2014, Ostrander 2010*). While the definition of the “optical redox ratio” is not standardized, FAD/(NADH+FAD) was used in the study *Georgakoudi I., 2012*, and multiple labs (for example, *Ostrander, J.H., et al. Cancer Res, 2010; Sepehr, R., et al. J Biomed Opt, 2012; Maleki, S., et al. J Biomed Opt, 2013; Palmer, S., et al, J Biophotonics, 2017; Lagarto, J.L., et al. Biomed Opt Express, 2018.*) use an alternate formula for the ratio, NAD(P)H/FAD or NAD(P)H/(NAD(P)H+FAD). Google Scholar searches for optical redox ratio “NADH/FAD” or “NADH/(NADH+FAD)” (plus other iterations) yielded ~1300 results and searches for optical redox ratio “FAD/NADH” or “FAD/(NADH+FAD)” (plus similar iterations) yielded ~670 results. Although this is not a comprehensive or accurate assessment of publications with the two definitions, it does demonstrate that it is common to have either NAD(P)H or FAD in the numerator, and by this measure, the NAD(P)H/FAD definition is more commonly used. Therefore, we justified the reason we used NAD(P)H/FAD in the Materials and Methods.

FROM MATERIALS AND METHODS: (P17) “The optical redox ratio is the relative fluorescence intensities of NAD(P)H and FAD and provides an optical measurement of the redox state of the cell⁵⁶⁻⁵⁸. Different optical redox ratio definitions with either NAD(P)H or FAD in the numerator can be found in the literature⁵⁶⁻⁵⁸. Here, we use NAD(P)H/FAD as the optical redox ratio.”

3) The originality of the approaches to clustering and machine-learning algorithm based on metabolic parameters used by the authors is not completely clear. So, in the work (Hao Zhou et al. “Non-invasive Optical Biomarkers Distinguish and Track the Metabolic Status of Single Hematopoietic Stem Cells”, *Cell Press, Volume 23, Issue 2, 2020*) were used the same approaches for the metabolism-based sorting of stem cells during differentiation. It would be helpful if the authors can include in the discussion more information about FLIM metabolic imaging in stem cell research and approaches to segmentation and machine-learning algorithm used by other researchers for stem cell metabolic sorting.

We thank the reviewer for the comment about adding more literature related to machine-learning applications for autofluorescence and stem cell differentiation. Hence, we added more related publications in the introduction.

FROM INTRODUCTION: (P3) “Several groups have demonstrated that autofluorescence lifetime imaging can non-invasively track stem cell metabolic activities in real time, including monitoring mesenchymal stem cell differentiation into adipocytes^{23, 24}, osteocytes^{24, 25}, and chondrocytes²⁵, distinguishing differentiation of hPSCs into dermal and epidermal lineages²⁶, metabolic difference between hPSCs and feeder cells²⁷, and hematopoietic stem cells at different stages²⁸. These prior studies indicate that OMI is suitable to detect the metabolic changes that occur during CM differentiation.”

FROM INTRODUCTION: (P4) “Compared to previous studies²³⁻²⁸, we specifically contribute a predictive model based on OMI to determine CM differentiation outcome as early as day 1.”

4) Could this methodology be adapted potentially for *in vivo* assessment of the early stage of stem cell differentiation using intravital imaging?

We appreciate the reviewer’s suggestion to discuss the potential of *in vivo* assessments of early stage stem cell differentiation. Cells at distinct differentiation stages exhibit altered metabolic states (*Front. Cell Dev. Biol.*, 2020, 8: 1-16). Autofluorescence imaging provides a label-free method to monitor these metabolic changes *in vivo*. Hence, OMI could be used for *in vivo* assessment of stem cell differentiation. We have added this potential application in the discussion.

FROM DISCUSSION: (P14) “A few studies have demonstrated that intravital autofluorescence imaging can monitor cellular metabolic activities, including tumor treatment response⁵⁰ and heterogeneity of tumor tissues⁵¹. Therefore, the methods described in the current study could be used to track stem cell differentiation *in vivo*, with applications in label-free assessment of stem cell-generated tissue integration for *in vivo* tissue repair.”

Reviewer #3 (Remarks to the Author):

The manuscript by Qian et al. describes an elegant biotechnology improvement for the *in-vitro* differentiation of human pluripotent cells into cardiomyocytes.

The investigators only vary the conditions at the beginning of the protocol (seeding density and concentration of Wnt pathway agonist). It is therefore not surprising that the authors find that varying those early conditions in the protocol will influence the outcome of the differentiation, but many other things may as well, or possibly even more so, which is not explored in this manuscript.

However, this investigation is really designed to investigate how early during the differentiation protocol one can measure whether the early conditions are

suitable for subsequent cardiomyocyte differentiation. While not introducing this research question at all well in the manuscript, the authors results are really spectacular: They can tell already after 24 hours (from metabolic information collected by measuring and imaging autofluorescence from many NAD(P)H and FAD molecules), and they can do this at a single-cell level. The discovery that successfully treated cells can be recognised in a recognisable Day1 cluster is unfortunately somehow buried in the supplemental information (currently Figure S3b).

We appreciate the review's comment about the day 1 UMAP clustering. We hence moved the representative UMAPs from supplemental into Figure 2. Updated Figure 2 below.

Figure 2. Multivariate analysis reveals unique metabolic profiles in cells differentiated into cardiomyocytes at day 1. (a) Uniform Manifold Approximation and Projection (UMAP) dimensionality reduction was performed on all 13 autofluorescence variables (optical redox ratio, NAD(P)H τ_m , τ_1 , τ_2 , α_1 , α_2 , and intensity; FAD τ_m , τ_1 , τ_2 , α_1 , α_2 , and intensity) for each cell and projected onto 2D space. Cells from all 11 conditions shown in Table 1 are plotted together. Data include cells from day 0, day 1, day 3, and day 5. Each dot represents one single cell, and $n = 25304, 25470, 26228,$ and 23484 cells for day 0, 1, 3, and 5, respectively. (b) Heatmap dendrogram clustering based on similarity of average Euclidean distances across all variable z-scores was performed on day 1 cells across all 15 conditions. Conditions are indicated by the CM differentiation efficiency percentages as noted by column labels at the top of the heatmap (quantified by flow cytometry cTnT+ on day 12, full conditions given in Table 1). Low differentiation efficiencies ($< 50\%$) are shaded in light gray and high differentiation efficiencies are shaded in dark gray ($\geq 50\%$). Z-score = $\frac{\mu_{\text{observed}} - \mu_{\text{row}}}{\sigma_{\text{row}}}$, where μ_{observed} is the mean value of each variable for each condition; μ_{row} is the mean value of each variable for all 15 conditions together, and σ_{row} is the standard deviation of each variable across all 15 conditions. Autofluorescence variables are indicated on the left side as row labels. $n = 30463$ cells from day 1. (c) Separated UMAP clusters for representative differentiation conditions. Conditions are labelled with original cell seeding density, CHIR99021 treatment concentration, and final cardiomyocyte differentiation efficiency quantified by flow cytometry (detailed in Table 1). $n = 13897, 13852, 4357,$ and 7601 cells for condition 65.5%, 51.8%, 19.6%, 15.1%, respectively.

While further discrimination between the 13 measurable variables in their data, the authors highlight that the NAD(P)H tau m variable alone is already somewhat accurate at predicting subsequent differentiation success (e.g. Fig.3c); however, what is for practical applications probably more important is that measuring just two variables NAD(P)H tau m and FAD tau m is pretty much as accurate as measuring all 13 (which is also consistent with what the authors then use in subsequent experiments, e.g. Fig.4).

We thank the reviewer for the comment about the practical implementation of this model. We agree with the reviewer that low and high differentiation efficiency conditions exhibit distinct NAD(P)H τ_m (Figure 1 and Figure 4). However, during revision we added 2 more cell lines and 4 conditions to improve the robustness of our model, and the new model indicates that NAD(P)H lifetime variables alone perform better than the combination of NAD(P)H and FAD τ_m (Fig. 3e). Given that all NAD(P)H lifetime variables must be measured to calculate NAD(P)H τ_m and that additional laser lines are needed to add FAD measurements, we conclude that NAD(P)H lifetime measurements provide a balance of good accuracy with reduced complexity. However, all NAD(P)H and FAD intensity and lifetime measurements provide the highest accuracy. We modified the results:

FROM RESULTS: (P9) “This model indicates that NAD(P)H lifetime variables alone perform better than the combination of NAD(P)H and FAD τ_m (Fig. 3e). Given that all NAD(P)H lifetime variables must be measured to calculate NAD(P)H τ_m and that additional laser lines are needed to add FAD measurements, we conclude that NAD(P)H lifetime measurements provide a balance of good accuracy with reduced complexity. However, all NAD(P)H and FAD intensity and lifetime measurements provide the highest accuracy.”

Minor Points:

- Consider providing a visual outline of the established differentiation protocol, highlighting the steps varied in this investigation and introducing this investigation’s focus on just studying varying early conditions in this protocol.

We thank the reviewer’s comment about outlining the established differentiation protocol. Hence, we added the differentiation protocol in Figure S1.

- Consider introducing this study's research question on how early the influence of these early conditions on subsequent differentiation success can be monitored

We thank the reviewer for their suggestion to highlight the influence of these early conditions on subsequent differentiation success. We have added these points in the introduction and discussion.

FROM INTRODUCTION: (P3) "These prior studies indicate that OMI is suitable to detect the metabolic changes that occur during CM differentiation. Early prediction of CM differentiation outcome can benefit CM manufacturing."

FROM DISCUSSION: (P13) "Our studies indicate that autofluorescence can predict CM differentiation efficiency at an early stage, which could enable real-time and/or in-line monitoring during cell manufacturing. This method could lower manufacturing costs and personnel time by flagging samples for timely interventions. Similar technologies could also impact other areas of regenerative cell manufacturing."

- Consider whether what is currently Fig S3 should be presented in the main manuscript to document the discovery of a reliable signal at Day1 already.

We appreciate the reviewer's suggestion. Hence, we moved part of Figure S3 into Figure 2.

Figure 2. Multivariate analysis reveals unique metabolic profiles in cells differentiated into cardiomyocytes at day 1. (a) Uniform Manifold Approximation and Projection (UMAP) dimensionality reduction was performed on all 13 autofluorescence variables (optical redox ratio, NAD(P)H τ_m , τ_1 , τ_2 , α_1 , α_2 , and intensity; FAD τ_m , τ_1 , τ_2 , α_1 , α_2 , and intensity) for each cell and projected onto 2D space. Cells from all 11 conditions shown in Table 1 are plotted together. Data include cells from day 0, day 1, day 3, and day 5. Each dot represents one single cell, and $n = 25304, 25470, 26228,$ and 23484 cells for day 0, 1, 3, and 5, respectively. **(b)** Heatmap dendrogram clustering based on similarity of average Euclidean distances across all variable z-scores was performed on day 1 cells across all 15 conditions. Conditions are indicated by the CM differentiation efficiency percentages as noted by column labels at the top of the heatmap (quantified by flow cytometry cTnT+ on day 12, full conditions given in Table 1). Low differentiation efficiencies ($< 50\%$) are shaded in light gray and high differentiation efficiencies are shaded in dark gray ($\geq 50\%$). $Z\text{-score} = \frac{\mu_{\text{observed}} - \mu_{\text{row}}}{\sigma_{\text{row}}}$, where μ_{observed} is the mean value of each variable for each condition; μ_{row} is the mean value of each variable for all 15 conditions together, and σ_{row} is the standard deviation of each variable across all 15 conditions. Autofluorescence variables are indicated on the left side as row labels. $n = 30463$ cells from day 1. **(c)** Separated UMAP clusters for representative differentiation conditions. Conditions are labelled with original cell seeding density, CHIR99021 treatment concentration, and final cardiomyocyte differentiation efficiency quantified by flow cytometry (detailed in Table 1). $n = 13897, 13852, 4357,$ and 7601 cells for condition 65.5%, 51.8%, 19.6%, 15.1%, respectively.

- Consider whether more cell lines should be tested.

We appreciate the reviewer's suggestion on adding more cell lines. We performed extra experiments with two more hPSC lines, including H13 ESCs and 19-9-11 iPSCs. A total of 4 new conditions were added into the dataset for the day 1 classification model performance evaluation, day 1 UMAP clustering, and z-score heatmap clustering. All the figures are updated for Figure 2, Figure 3, Figure S3, and Figure S4.

- Consider whether the clustering in Fig.2b is influenced by the nature of the cell line (H9 ESC vs IMR90 iPSC) as much as by the differentiation efficiency

We appreciate the reviewer's comment. We added two more hPSC lines and four more conditions into the experiments. The z-score heatmap was updated in Figure 2b. From the z-score heatmap, it seems that the majority of high and low differentiation efficiency conditions clustered separately except the 61.0% differentiation efficiency from the 19-9-11 iPSC line. This could be due to multiple factors, including cell seeding density and CHIR99021 concentration. We modified the results and figure (see above updated Figure 2).

FROM RESULTS: (P7) "Heatmap dendrogram clustering based on OMI variable z-scores revealed that cells under high differentiation efficiency conditions on day 1 were clustered closely together and distinct from cells under low differentiation efficiency conditions on day 1 (Figure 2b) except the 61.0% differentiation efficiency with 6 μ M CHIR treatment for 19-9-11 hPSCs. This outlier could be due to multiple reasons, e.g., the low CHIR99021 concentration or low cell seeding density. In the future, more variables, such as cell morphology, might be introduced to increase linear clustering performance."

- Consider whether abbreviations such as ROC and AUC should again be introduced in the result section (these are introduced in Introduction and Figure legend).

We thank the reviewer for their suggestion about introducing the abbreviation of ROC and AUC. We introduce ROC and AUC again in the results.

FROM RESULTS: (P8) "Performance of classifiers (receiver operating characteristic (ROC) area under the curve (AUC)) was evaluated on dataset 2."

- Consider whether panel lettering in current Figs 1 and 4 should be more comparable.

We thank the reviewer for suggesting comparable lettering in Figs 1 and 4. We have updated panel lettering of Figure 1.

Figure 1

- Consider whether current Figure 4 (experiments with NKX2.5EGFP) could be presented as suppl. material, or whether focus on measuring metabolic variables at Day8 should be better introduced and relationship with cardiac-like fibroblast differentiation better explored and possibly documented within these results.

We appreciate the reviewer's comment about day 8 NKX2.5-EGFP data. The reporter line NKX2.5^{EGFP/+} hPSCs allowed us to distinguish CMs from non-CMs. These non-CMs are likely fibroblasts and exhibit different metabolic preferences than fetal/newborn CMs that are more reliant on glycolytic activities (*J. Cardiovasc. Pharmacol.* 2010, 56(2): 130-140). This day 8 data further confirms that OMI can distinguish these metabolic differences between CMs and non-CMs at different stages. Although the central goal of the study was to predict CM differentiation efficiency at an early stage, this day 8 data provides supporting evidence that OMI measures meaningful differences between CMs and non-CMs at later stages. We have justified the significance of day 8 OMI data in the results and clarified the significance of distinguishing CMs from non-CMs on day 8 in the discussion.

FROM RESULTS: (P11) "OMI differences between CMs and non-CMs on day 8 further confirmed that autofluorescence can identify CMs at different stages during differentiation."

FROM DISCUSSION: (P12) "OMI distinguished different cell populations at multiple time-points in this differentiation protocol, with larger differences between low and high differentiation efficiencies on day 8 than on day 1. Day 1 cells are mainly primitive streak whereas cells on day 8 are at the end of differentiation and exhibit the glycolytic activities of fetal/newborn CMs⁴⁵. Therefore, larger differences are expected in OMI between CMs and non-CMs at day 8 compared to low and high differentiation efficiencies on day 1. OMI differences between low and high differentiation efficiencies at multiple days in this protocol indicates that this technology could continuously monitor stem cell differentiation stages."

- Consider whether investigating how CHIR99021 influences the altered metabolic indicators could be relevant (e.g. is the altered metabolism directly caused by Wnt signalling (in parallel with mesoderm induction) or downstream of successful mesoderm induction?)

We thank the reviewer about the comments on the cause of altered metabolic activities. We do not know clearly whether the metabolic switch was due to Wnt activation or mesoderm induction. We have tested Wnt activation by applying CHIR99021 at 0, 1, 2, 4 μ M to stem cell-derived endothelial cells. The results did not indicate consistent OMI changes upon CHIR99021 treatment (FIG. D). CHIR99021 also has been shown as a molecule to boost cell proliferation (Joseph Wu, Cell stem cell, 2020), so it is not clear whether the OMI changes were due to Wnt activation or induction of cell proliferation in the experiment below. Previous studies have shown that Wnt signaling and glycolytic activities are linked during hPSC differentiation into mesoderm. Therefore, we highlighted that statement in the discussion.

FROM DISCUSSION: (P11) "Recent evidence links Wnt signaling and glycolytic activities during hPSC differentiation into mesoderm ^{39, 40}."

Reviewers' Comments:

Reviewer #1:

Remarks to the Author:

The authors have sufficiently addressed all comments.